# Neural Cues and Genomic Clues: NGS Insights into Neurogenic Sarcopenia and Muscle Atrophy

**DOI:** 10.3390/ijms262211185

**Published:** 2025-11-19

**Authors:** Darya Kupriyanova, Airat Bilyalov, Nikita Filatov, Sergei Brovkin, Dmitrii Shestakov, Natalia Bodunova, Oleg Gusev

**Affiliations:** 1Federal Center of Brain Research and Neurotechnology of the Federal Medical Biological Agency (FMBA) of Russia, 117513 Moscow, Russia; 2SBHI Moscow Clinical Scientific Center Named After Loginov MHD, 111123 Moscow, Russia; 3Life Improvement by Future Technologies (LIFT) Center, 121205 Moscow, Russia; 4National Medical Research Centre of Cardiology Named After Academician E. I. Chazov, 121552 Moscow, Russia; 5Intractable Disease Research Center, Graduate School of Medicine, Juntendo University, Tokyo 113-8421, Japan

**Keywords:** sarcopenia, neuromuscular disease, aging, neuromuscular junction (NMJ), skeletal muscle, motor neuron, synapse, NGS, omics data, model, artificial intelligence

## Abstract

Sarcopenia is a progressive loss of skeletal muscle mass and strength with major clinical and economic consequences. While traditional models emphasize mitochondrial dysfunction, inflammation, and proteostasis imbalance, emerging data highlight a neurogenic component involving motor neuron loss, fiber denervation, neuromuscular junction remodeling, and disrupted trophic signaling. To synthesize current evidence on neurogenic mechanisms of sarcopenia revealed by next-generation sequencing and related multi-omics, to map molecular networks across cell types, and to outline translational opportunities for diagnostics and targeted therapy. A narrative review of human and animal studies indexed in PubMed, Web of Science, and Scopus through November 2025. Search terms combined sarcopenia, denervation, neuromuscular junction, neurotrophic signaling, genomics, transcriptomics, epigenomics, single-cell, and spatial transcriptomics. Eligible studies reported omics or physiological endpoints related to neuromuscular function. Convergent omics data support a central role of the nervous system in the onset and progression of sarcopenia. Genetic and regulatory factors linked to denervation, transcriptomic signatures of junctional disassembly, and cell-specific dysfunctions in motor neurons, Schwann cells, satellite cells, and fibro-adipogenic progenitors have been identified. Epigenetic and transcriptional networks underlying neuromuscular homeostasis, along with candidate circulating biomarkers, provide targets for clinical translation. Neurogenic sarcopenia represents a tractable target for precision prevention and therapy. Integration of multi-omics, artificial intelligence, and advanced models such as innervated organoids and NMJ-on-chip systems can accelerate target validation and enable personalized strategies to preserve neuromuscular function.

## 1. Introduction

Sarcopenia is a pathological condition characterized by progressive loss of muscle mass and strength. The term sarcopenia was introduced by Rosenberg in 1989 to denote age-related loss of skeletal muscle mass [1], but it was only in 2016 that sarcopenia was included as a disease in the International Classification of Diseases (ICD-10) [2]. Due to the EWGSOP2 consensus update in 2018, sarcopenia is no longer considered merely a geriatric syndrome but an independent disease of skeletal muscles that may not be associated with aging, as it also occurs in young people [3]. Moreover, sarcopenia is an important prognostic criterion for survival and clinical complications in patients with oncological pathologies, kidney dysfunction, liver diseases, and metabolic disorders [4].

Findings from the GLIS working group (The Global Leadership Initiative in Sarcopenia) demonstrate a broad spectrum of consequences of sarcopenia: from increased mortality and reduced quality of life to an elevated risk of falls, fractures, hospitalizations, and reduced life expectancy [5,6,7,8]. This evidence supports the notion that sarcopenia not only leads to increased direct medical costs, placing a significant financial burden on patients’ families and healthcare systems, but also impacts the national economy as a whole. The loss of muscle mass and strength results in reduced functional activity, loss of working capacity, and early retirement due to health reasons. This leads to a reduction in the labor potential of the population and decreased productivity, which negatively affects the macroeconomic indicators of the state [9].

Traditional models of sarcopenia pathogenesis focus on mitochondrial dysfunction and oxidative stress [10], chronic inflammation (“inflammaging”) [11], metabolic disturbances [12], and proteostasis imbalance [13] as key factors in skeletal muscle atrophy. However, accumulated data indicate a critical role of the neuron–muscle axis in the initiation and progression of this pathological condition. Age-related loss of motor neurons leads to denervation of muscle fibers and impaired reinnervation [14]. Concurrently, remodeling of neuromuscular junctions (NMJs) occurs, with dysregulation of the Agrin-LRP4-MuSK signaling cascade [15] and altered expression of neurotrophins and their receptors [16], which disrupts signaling along the neuron–muscle axis. These processes form the basis of the neurogenic hypothesis of sarcopenia, where the primary trigger of muscle failure is localized in the central nervous system.

Despite progress, fundamental gaps remain in understanding the neurogenic mechanisms of sarcopenia. The central problem boils down to the causal relationship: is motor neuron degeneration the initiating factor or a consequence of the loss of trophic signals from the atrophying muscle? The heterogeneity of clinical phenotypes indicates the existence of unidentified modifying factors. Finally, the full spectrum of molecular-genetic signals mediating bidirectional communication between neuron and muscle in sarcopenia has not been determined.

A systematic review of electromyography data supports the notion that early NMJ dysfunction is an early and significant predictor of sarcopenia development. Key changes include instability of synaptic transmission, loss of motor units with compensatory remodeling, and impaired neuromuscular activation that precede muscle atrophy. Electromyographic parameters combined with biomarkers (C-terminal agrin fragment, neurofilament light chain) represent sensitive indicators of early NMJ degradation [17]. However, traditional methods do not account for the molecular-genetic background of patients with sarcopenia, and the commonly used real-time PCR-based assessment of expression levels only covers pre-selected candidate genes.

Next-generation sequencing (NGS) offers fundamentally new opportunities for establishing the role of the nervous system in the initiation and progression of sarcopenia [18]. Technical progress in recent years and a substantial reduction in the cost of NGS technologies have facilitated the widespread adoption of genomic analysis methods, including gene expression assessment, chromatin accessibility profiling, DNA methylation pattern analysis, and whole-genome sequencing [19] A critical achievement has been the adaptation of these approaches for single-cell analysis, which has enabled the detailed examination of cell-specific mechanisms of pathological processes. The application of NGS technologies in sarcopenia research demonstrates their high efficacy in identifying molecular mechanisms of neuromuscular dysfunction [18].

Whole-exome sequencing (WES) and whole-genome sequencing (WGS) reveal both common and rare genetic variants in key genes involved in the regulation of synaptic transmission and neuroinflammation associated with accelerated denervation [20,21]. Bulk RNA sequencing (bulk RNA-seq) demonstrates systemic dysregulation of signaling cascades in the neuromuscular axis, characterized by altered expression of neurotrophins and their receptors, activation of proteolytic systems, and induction of denervation-associated genes in skeletal muscle [22].

Single-cell RNA sequencing (scRNA-seq) technology allows for the identification of dysfunction in specific cellular subpopulations, including impaired satellite cell activity and pathological proliferation of fibro-adipogenic progenitors in response to age-related stress [23]. Integration of chromatin accessibility data (ATAC-seq, ChIP-seq) with methylation profiles (WGBS) enables the reconstruction of regulatory networks controlling the expression of genes involved in neuromuscular homeostasis under conditions of sarcopenia development [19].

The comprehensive application of these multi-level approaches, schematically presented in Figure 1, allows for the integration of disparate data into a holistic multi-omics framework that reveals the neurogenic component of sarcopenia pathogenesis.

Although metabolic, inflammatory, and endocrine mechanisms in sarcopenia have been extensively studied, the neurogenic dimension remains comparatively underexplored. Existing reviews often address neuromuscular junction (NMJ) remodeling or denervation-related signaling as isolated phenomena, without integrating multi-omics insights or linking findings to translational applications. This review seeks to systematically examine and synthesize current evidence from next-generation sequencing (NGS) and other high-throughput approaches to clarify the structural and functional contributions of the nervous system to sarcopenia pathogenesis. We place particular emphasis on molecular pathways driving disease onset and progression, as revealed through transcriptomic, epigenomic, and proteomic analyses. In doing so, we highlight how integrative omics and advanced experimental platforms—such as human neuromuscular organoids and in vivo modeling—can help close the gap between mechanistic discovery and clinical translation.

## 2. Methods

### 2.1. Search Strategy and Selection Criteria

We conducted a narrative review focused on neurogenic mechanisms contributing to sarcopenia, with an emphasis on findings derived from omics technologies. A comprehensive search was performed across PubMed, Web of Science, and Scopus from database inception through November 2025. Search terms combined controlled vocabulary and free-text keywords. Core concepts included: sarcopenia, denervation, neuromuscular junction, motor unit, neurotrophin, BDNF, TrkB, IGF-1, irisin, mitochondrial biogenesis, PGC-1α, genomics, transcriptomics, epigenomics, single-cell, spatial transcriptomics, RNA-seq, ATAC-seq, ChIP-seq, WES, and WGS. Boolean operators were applied using AND within concepts and OR across synonyms. Reference lists of eligible studies and key reviews were hand-searched to identify additional relevant publications.

### 2.2. Eligibility Criteria

We included peer-reviewed human or animal studies reporting molecular or physiological outcomes related to innervation status or neuromuscular function in the context of sarcopenia or aging skeletal muscle. Eligible studies encompassed original omics datasets and mechanistic investigations elucidating neurogenic pathways. Exclusion criteria were: non-English publications, conference abstracts lacking full datasets, narrative reviews without primary data, and studies not reporting neuromuscular or innervation-related outcomes.

### 2.3. Data Extraction and Synthesis

From each study, we extracted the following: model system (human or animal), species, cohort characteristics, tissue or cell type analyzed, assay platform, primary endpoints, key genes and pathways, and main findings pertaining to neurogenic mechanisms. Due to heterogeneity in study designs and outcomes, we synthesized the data narratively. Results were organized by omics modality and biological theme, including genetic variation, bulk and single-cell transcriptomics, chromatin accessibility and DNA methylation, and candidate translational biomarkers.

### 2.4. Risk of Bias and Limitations

As this was a narrative review, we did not conduct meta-analysis or apply formal risk of bias assessment tools. We recognize potential limitations, including publication and selection bias, heterogeneity in sarcopenia definitions, and variability in pre-analytical handling across omics platforms. These factors were carefully considered in our interpretation of the consistency, robustness, and translational relevance of the findings.

## 3. The Neurogenic Paradigm in Sarcopenia Pathogenesis: From Synaptic Dysfunction to Systemic Neurodegeneration

This section of the review provides a comprehensive analysis of the motor unit—the fundamental structure comprising an alpha-motoneuron and the ensemble of muscle fibers it innervates. Primary focus is placed on a comparative analysis of the physiological state of motor unit components and their pathological transformation in sarcopenia. The morphofunctional characteristics of alpha-motoneurons are examined in detail, including their age-associated changes, mechanisms of neuromuscular transmission, and the molecular and electrophysiological disturbances associated with degenerative processes. Particular emphasis is given to the role of the neuromuscular junction (NMJ) as a key element of the neuromuscular axis, analyzing its structural components, mechanisms of age-related destabilization, and the significance of glial support. The molecular underpinnings of neurogenic sarcopenia, including the neurotrophic factor system and mechanisms of muscle fiber denervation, are explored. The section concludes by analyzing the place of sarcopenia within the context of neurodegenerative diseases, thereby revealing common pathogenetic mechanisms and potential targets for therapeutic strategies.

### 3.1. Architecture of the Neuromuscular Axis and Its Remodeling in Aging and Sarcopenia

Alpha-motoneurons are specialized spinal cord neurons that innervate skeletal muscle fibers and form the basis of motor units. These neurons exhibit significant heterogeneity in size and functional characteristics depending on the type of muscle fibers they innervate. Small motoneurons primarily control slow oxidative fibers (type I), which are responsible for postural control and endurance, while large motoneurons innervate fast glycolytic fibers (type IIx/IIb), which generate maximal force and power [24]. The aging process is accompanied by a progressive loss of motoneurons, which becomes particularly pronounced after the age of 60. Morphometric studies of human tissue demonstrate a reduction in motoneuron numbers of up to 50% in individuals over 60 compared to early and middle adulthood [25]. Stereological studies in animal models corroborate these findings; 24-month-old F344 rats show an approximately 20% decrease in hypoglossal motoneurons compared to young animals [26]. Large, fast-twitch motoneurons innervating type II fibers appear especially vulnerable to age-related changes. This selective degeneration explains the preferential loss of muscle strength and power with aging, a hallmark feature of sarcopenia [27]. Data from surface electromyography suggest that age-related functional changes are best explained by a combined loss of approximately 40% of motor units coupled with a ~50% reduction in the number of fast-twitch muscle fibers [28]. In addition to numerical decline, surviving motoneurons undergo significant morphological alterations. The size of motoneuron cell bodies decreases by approximately 25% in aged animals, which is accompanied by a 20% reduction in muscle strength and atrophy of fast-twitch fibers [26]. Figure 2 schematically illustrates the described structural and functional changes in the motoneuron and NMJ that occur during aging.

At the molecular level, aging motoneurons exhibit critical impairments in proteostasis and autophagy. Markers of the autophagic pathway, LC3 and p62, show significant accumulation in lumbar motoneurons: LC3 levels increase by approximately 150% at 24 months of age compared to 6 months, while p62 increases by 14% [29]. These changes indicate impaired protein degradation, leading to the accumulation of damaged organelles and protein aggregates, and a reduction in the overall metabolic activity of motoneurons.

Motoneurons are particularly sensitive to oxidative stress due to their high metabolic activity and large cell size. Experimental data from mice with motoneuron-specific knockout of the Sod1 (superoxide dismutase-1) gene demonstrate accelerated age-related denervation of NMJs, a decrease in the number of large axons, and an increase in the proportion of small axons compared to wild-type mice of the same age [30]. These results confirm the critical role of antioxidant defense in maintaining the structural integrity of motoneurons.

Age-related changes in motoneurons also affect their electrophysiological properties. The intrinsic excitability of motoneurons decreases in older adults, manifested by a reduction in persistent inward currents (PICs) and common synaptic inputs to motor units—the groups of muscle fibers innervated by a single motoneuron [31]. These changes directly impact the ability to generate and sustain muscle activity. Despite these pronounced pathological changes, the neuromuscular system activates compensatory mechanisms, including an increase in the number of muscle fibers per motor unit [32] and molecular adaptations [33]. Therapeutic approaches, particularly functional electrical stimulation, demonstrate efficacy in slowing age-related declines [34].

Impairment of NMJ integrity is manifested by an increased number of NCAM-positive denervated fibers [35], fragmentation of acetylcholine receptor (AChR) clusters, and accelerated degradation of key synaptic proteins such as LRP4 [15]. In response to denervation, surviving motoneurons initiate a process of collateral sprouting—the formation of new terminal branches to reinnervate denervated muscle fibers [36,37]. However, this compensatory reinnervation has significant limitations: not all fibers are successfully reinnervated, the quality of newly formed NMJs is often reduced, and the regenerative capacity of the nervous system progressively declines with age [38]. Nevertheless, the nervous system retains significant plasticity even in advanced age, opening prospects for developing effective therapeutic strategies.

Motoneurons are the primary source of synthesis and secretion of agrin (Agrn)—a key regulatory molecule essential for the formation and maintenance of the structural and functional integrity of the NMJ [39,40,41]. Neuronal agrin is transported along the axon to the nerve terminals and released into the synaptic cleft, where it binds to the LRP4 receptor on the postsynaptic muscle membrane [42,43]. Experimental studies in model animals show that disruption of the axonal transport of Agrn mRNA leads to severe defects in synaptic transmission and significant motor impairments [44]. The critical role of the motoneuronal source of agrin is supported by clinical cases of congenital myasthenic syndromes with mutations in the AGRN gene, demonstrating the indispensability of motoneuronal agrin for synapse formation and survival [45]. Agrin binding to the LRP4 receptor activates the tyrosine kinase MuSK. In complex with DOK7, this signaling cascade induces the aggregation and stabilization of postsynaptic acetylcholine receptors (AChRs) and promotes the formation of postsynaptic folds, which significantly enhance the efficiency of neuromuscular transmission [46]. In experimental models in aged animals, administration of the ActR-Fc-nLG3 complex, containing the C-terminal domain of agrin, enhanced AChR clustering, increased the complexity and folding of the postsynaptic surface, improved NMJ stability, and enhanced muscle endurance. Thus, agrin can directly strengthen the postsynaptic architecture and improve the functional efficiency of the neuromuscular synapse [47]. With age, agrin levels in skeletal muscles progressively decrease. Experimental removal of agrin from muscle tissue induces premature signs of muscle aging and the development of sarcopenia. Conversely, increasing local agrin levels (via a mini-agrin form) improves muscle function, partly through the activation of Yap signaling and increased expression of dystroglycan, linking the molecular role of agrin to the maintenance of muscle mass and structural integrity during aging [48]. Clinical studies show that genetic variants in the AGRN gene are associated with measures of muscle mass, strength, and the concentration of the C-terminal agrin fragment (CAF) in plasma. Altered CAF levels are considered potential biomarkers of NMJ instability in aging, reinforcing the concept of agrin’s role in the pathogenesis of human sarcopenia [49,50].

The neuromuscular junction (NMJ) is a highly specialized structure comprising three key elements: the presynaptic nerve terminal, the postsynaptic membrane of the muscle fiber, and the surrounding perisynaptic Schwann cells. It ensures efficient transmission of the nerve impulse from the motoneuron to the muscle fiber [51]. The presynaptic component includes the terminal part of the motoneuron axon, which contains a high concentration of synaptic vesicles filled with acetylcholine (ACh). The active zones of the presynaptic membrane are characterized by clusters of voltage-gated calcium channels and specialized proteins (RIM, Bassoon) that mediate calcium-dependent exocytosis of the neurotransmitter [52]. The density and morphological organization of these active zones are critical determinants of ACh release efficiency. The signaling molecule agrin, which activates the LRP4-MuSK cascade on the postsynaptic membrane, plays a key role in synapse organization [53]. This signaling ensures the clustering and maintenance of nicotinic acetylcholine receptors (nAChRs) in the postsynaptic region. The postsynaptic components are characterized by deep invaginations of the muscle membrane, forming junctional folds that significantly increase the surface area [54]. The endplate region exhibits an exceptionally high density of nAChRs (up to 10,000 receptors per µm^2^), organized into dense functional clusters [55]. With age, NMJ morphology undergoes significant pathological changes characterized by progressive destabilization of the synaptic architecture. The primary signs of age-related alteration include fragmentation of AChR clusters and a reduction in the area of overlap between the presynaptic nerve terminal and the postsynaptic endplate. Morphometric studies demonstrate a strong correlation between the degree of pre-/postsynaptic overlap and the efficiency of synaptic transmission. Importantly, physical exercise can partially prevent this structural disintegration and functional decline [56]. The structural changes in the aging NMJ lead to impairments in synaptic transmission at multiple levels. Presynaptic impairments include a reduced density of active zones and diminished calcium influx, resulting in decreased endplate potential. Postsynaptic impairments are characterized by a reduction in nAChR density and altered receptor kinetics, leading to a decrease in the amplitude and a slowing of the endplate potential kinetics [57].

Perisynaptic Schwann cells (PSCs) play a critical role in maintaining the structural and functional stability of the NMJ [58]. These specialized glial cells envelop the synaptic area and perform multiple functions: trophic support via the secretion of neurotrophic factors, modulation of synaptic activity, and phagocytic removal of degenerating synaptic components [59,60,61]. During aging, PSCs undergo phenotypic changes characterized by the activation of pro-inflammatory programs and a reduction in trophic support. These changes contribute to NMJ destabilization and impair reinnervation processes [60].

### 3.2. Molecular Determinants of Neurogenic Muscle Atrophy: Neurotrophic Factors and Signaling Cascades

#### 3.2.1. Brain-Derived Neurotrophic Factor (BDNF) in the Maintenance of NMJ Structure and Function

A key molecular determinant of NMJ status and muscle plasticity is BDNF (Brain-Derived Neurotrophic Factor). This neurotrophin is produced both in the central nervous system and in skeletal muscle, where it functions as a myokine, influencing muscle plasticity and regeneration in response to contraction and physical exercise [62]. Its role is particularly relevant in the context of sarcopenia [63]. Skeletal muscle expresses and secretes BDNF, especially during physical activity, with different forms of the protein—mature BDNF and pro-BDNF—having different receptor targets and exerting opposing effects on NMJ regulation [62,63].

The molecular mechanisms of BDNF action in muscle involve several key processes relevant to sarcopenia, including the regulation of satellite cell activity [64], myogenesis [65], and mitochondrial biogenesis [66]. Key mechanisms include its influence on regeneration: the absence of BDNF, produced by both skeletal muscle and satellite cells, leads to reduced Pax7 expression (a satellite cell marker), slowed proliferation and differentiation of myoblasts, and delayed muscle recovery, while restoring BDNF levels mitigates these defects [64]. BDNF is also involved in maintaining NMJ integrity by regulating presynaptic acetylcholine release and postsynaptic structure, which is critical for preserving motor units with age [63].

The BDNF signaling pathways mediating these effects involve various receptors and intracellular cascades. Mature BDNF activates the TrkB receptor, which enhances PI3K/Akt and MAPK signaling, supporting cell survival and anabolic processes [67]. In contrast, pro-BDNF can bind to the p75NTR receptor and activate pro-inflammatory NF-κB/JNK pathways, promoting inflammation and motoneuron apoptosis, thereby negatively impacting synaptic transmission [67,68,69]. Another key mechanism is metabolic reprogramming via PPARδ (Peroxisome Proliferator-Activated Receptor Delta)—a transcriptional factor regulating lipid metabolism. Muscle-produced BDNF induces PPARδ expression, initiating a program that regulates lipid metabolism and post-exercise recovery; conversely, muscle BDNF deficiency disrupts β-oxidation and mitochondrial dynamics [70].

The assessment of BDNF as a clinical biomarker and therapeutic target is currently limited. Cohort data, for example, from the Korean Frailty and Aging Cohort Study (KFACS), indicate that plasma BDNF levels may reflect “muscle quality” (MQ–the amount of force and/or power per unit of muscle mass) [71] rather than simply its quantity, and perform better in tandem with other biomarkers than alone [72]. Although interventional studies with physical exercise in animal models show that activity and immobilization alter serum and muscle BDNF levels [62,73], there are no direct randomized data from human cohorts on BDNF modulation in sarcopenia.

#### 3.2.2. Neurotrophin-4 (NT-4) and TrkB Signaling in Maintaining Neuromuscular Synapse Stability

Similar to BDNF, Neurotrophin-4 (NT-4) exerts its biological effects in muscle tissue through Tropomyosin receptor kinase B (TrkB) receptors and the p75NTR receptor, localized on motor axons, muscle fibers, and Schwann cells. This distributed expression enables retrograde and paracrine trophic communication within the NMJ [33,74].

A key aspect of synaptic stability regulation is the BDNF/NT-4 ratio, as well as the balance between the full-length TrkB-FL receptor isoform and the truncated TrkB-T1 isoform. Age-related changes are characterized by a shift in the TrkB-FL/TrkB-T1 balance towards an increase in the truncated isoform and increased p75NTR expression, which correlates with impairments in synaptic transmission during aging. NT-4/TrkB signaling influences the phosphorylation of key exocytosis proteins Munc18-1 and SNAP-25 via activation of PKC-dependent cascades, directly regulating acetylcholine release at the NMJ [33]. Activation of TrkB receptors by neurotrophins links them to classical pathways mediating cell survival, anabolism, and synaptic plasticity, making TrkB signaling critically important for maintaining neuromuscular complex function [16]. Signaling via NT-4/TrkB supports NMJ stability and indirectly influences muscle fiber size and strength by preserving effective innervation and synaptic transmission [75,76]. Mice with constitutively reduced TrkB expression exhibit fragmentation of postsynaptic ACh receptor areas, increased frequency of impaired neuromuscular transmission, and reduced muscle strength and fiber area, reproducing characteristic features of muscle aging [76].

Data on NT-4 level dynamics under various conditions remain contradictory. In a number of experiments, NT-4 levels remained unchanged during short-term unloading (a form of subtotal disuse) [77,78], whereas TrkB receptor expression decreased [78]. Delivery of NT-4 to the aging muscle of aged rats improved morphological and functional parameters in preclinical models, making NT-4 a promising candidate for therapeutic strategies in sarcopenia [79]. However, the preclinical positive effects of NT-4 require evaluation of safety, pharmacokinetics, and efficacy in humans; available studies currently lack large-scale clinical trials.

#### 3.2.3. NGF and NT-3: Additional Regulators of the Neuromuscular Axis Neuritrophin-3 (NT-3)

NT-3 plays an important role in preserving and restoring neuromuscular connections during aging and after injury. Through activation of its high-affinity receptor TrkC, it stimulates axonal growth and sprouting, including intracellular signaling cascades that promote axon regeneration. Administration of NT-3 improves sensorimotor recovery after stroke by increasing functional connectivity between the nervous and muscular systems [80]. An important aspect of NT-3 action is its support of Schwann cells via activation of the TrkC/ERK/c-Jun pathway, which maintains their reparative phenotype after denervation and improves axon conduction and directed regeneration, promoting the restoration of innervation [81]. NT-3 also demonstrates pronounced protective effects on muscle cells, reducing caspase-3 expression, thereby decreasing myocyte apoptosis after nerve injury, increasing sarcoplasmic Ca^2+^-ATPase content, and supporting calcium homeostasis, which is critical for contractile function [82]. Gene therapy using AAV-NT-3 showed pronounced anabolic effects, including an increase in muscle fiber diameter via activation of mTOR-dependent pathways. The effect was most pronounced in the population of fast-twitch glycolytic fibers [83].

#### 3.2.4. Nerve Growth Factor (NGF)

NGF demonstrates pronounced anti-catabolic effects, especially under conditions of metabolic stress. In models of obesity and type 2 diabetes, NGF treatment led to reduced levels of myostatin—a key inhibitor of muscle growth—as well as reduced markers of autophagy and ubiquitin-ligase activity, indicating a decrease in catabolic processes and contributing to the preservation of muscle mass in metabolic disorders. NGF stimulates anabolic signals, increasing Cyclin D1 levels, indicative of cell cycle activation and regenerative processes, improves protein synthesis in myocytes, and supports the regenerative potential of muscle tissue through the activation of proliferative signals [84]. An important component of NGF action is its high efficacy as a neuroprotectant, promoting motoneuron survival in a model of denervated motoneurons innervating extraocular muscles [85].

#### 3.2.5. GDNF and Its Role in Supporting the Neuromuscular System

GDNF is a neurotrophic factor from the transforming growth factor-β (TGF-β) family, first identified as a potent survival factor for dopaminergic neurons [86,87]. This protein is expressed in various tissues, including the central nervous system, peripheral nerves, and skeletal muscles [88,89], which is particularly important in the context of sarcopenia. GDNF exerts its biological effects through a complex receptor system consisting of GFR-α (GDNF family receptor α) and the receptor tyrosine kinase RET [90]. This signaling pathway provides both retrograde signal transmission from muscle to motoneurons in the spinal cord and local paracrine effects in the region of neuromuscular junctions [91,92]. Immunohistochemical studies show that GDNF is concentrated near the plasma membranes of skeletal muscles, indicating its local role in the formation and maintenance of the NMJ [93]. This localization makes GDNF a key component of the signaling network that supports the innervation of muscle fibers and the functional connectivity of the NMJ.

The mechanisms of GDNF influence on the neuromuscular system include support of motoneurons and modulation of neuromuscular junctions. Experimental data demonstrate that GDNF has a powerful neuroprotective effect on α-motoneurons in the spinal cord. Studies in animal models have shown that exogenous administration of GDNF into atrophied muscle can partially restore the population of α-motoneurons, indicating a retrograde trophic effect from muscle to spinal cord [91]. During aging and the development of sarcopenia, significant changes occur in GDNF expression. In the spinal cord and peripheral nerves of aging animals, a decrease in the expression of GDNF and its receptors is also observed, along with reduced levels of other important neurotrophic factors such as neuregulin-1 [91,92], confirming the systemic nature of age-related changes in GDNF-dependent signaling pathways. In patients with neurodegenerative diseases, low GDNF levels correlate with signs of sarcopenia and partially normalize after rehabilitation, underscoring its significance as a potential biomarker [94]. However, under conditions of immobilization or simulated microgravity, changes in plasma GDNF levels are not always observed [95], highlighting the difference between local regulation and systemic indicators.

#### 3.2.6. IGF-1 as a Key Anabolic Factor and Potential Biomarker of Sarcopenia

Insulin-like Growth Factor-1 (IGF-1) is a key anabolic factor playing a central role in regulating the growth, development, and maintenance of muscle tissue throughout human life [96]. In the body, IGF-1 is synthesized primarily in the liver under the control of growth hormone but is also produced directly in muscles as a myokine, influencing satellite cells and muscle tissue remodeling processes [97]. In addition to regulating protein synthesis and repair, IGF-1 modulates vascular support of tissues, making it a multi-system regulator of the functions of aging organs [98]. The molecular mechanisms of IGF-1 action on skeletal muscle are well studied and include the activation of receptor-dependent intracellular cascades controlling the balance between protein synthesis and breakdown in myocytes and satellite cells. Activation of the IGF-1 receptor initiates the PI3K-Akt-mTORC1 cascade, leading to enhanced protein translation and stimulation of muscle fiber growth. This pathway is the primary mechanism of the anabolic action of IGF-1 in skeletal muscle [99,100].

IGF-1 acquires particular importance in the context of neuromuscular integration, exerting a neurotropic effect on motoneurons and the NMJ. After denervation, local delivery of IGF-1 reduces muscle atrophy and preserves strength in experimental models [101]. Of special significance is the expression of the muscle-specific isoform mIGF-1, which stabilizes muscle structure and improves recovery, which is important for neuronal-muscular communication [102,103]. Local expression of mIGF-1 reduces the activity of caspases and components of the ubiquitin–proteasome system, contributing to the preservation of muscle and neuronal tissue [103].

A large cross-sectional study among older adults (n ≈ 3276) demonstrated significant differences in IGF-1 levels between the group with sarcopenia and the healthy control group. IGF-1 was identified as an independent marker associated with the Appendicular Skeletal Muscle Mass Index (ASMI) after adjustment for various covariates [104]. A systematic review of biomarkers in patients with hip fractures identified decreased IGF-1 concentrations as one of the characteristic changes in individuals with sarcopenia, making IGF-1 a potential component of biomarker panels for the clinical assessment of muscle function [105].

Despite convincing population data, evidence for the effectiveness of systemic IGF-1 therapy in age-related sarcopenia is still insufficient. This underscores the need for further clinical research aimed at developing therapeutic strategies based on the modulation of IGF-1 signaling in sarcopenia.

#### 3.2.7. Regulatory Role of Irisin in Muscle Homeostasis and Sarcopenia Development

Irisin is a myokine produced by the proteolytic cleavage of the transmembrane precursor protein FNDC5 (Fibronectin type III domain-containing protein 5) and is induced by physical exercise [106]. FNDC5 is expressed predominantly in skeletal muscle and adipose tissue, where its expression is regulated by the transcriptional coactivator PGC-1α. The process of irisin formation involves the synthesis of the membrane protein FNDC5 in response to PGC-1α activation, proteolytic cleavage of FNDC5, releasing the N-terminal domain, and secretion of irisin into the bloodstream, where it acts as an endocrine factor. The expression of FNDC5 and subsequent production of irisin are closely linked to physical activity: aerobic exercise and resistance training stimulate PGC-1α expression, leading to increased FNDC5 levels in muscle and elevated circulating irisin, making it an important mediator of the beneficial effects of exercise on the body [107]. Expression levels of irisin, as well as its precursor Fndc5, decrease at both the mRNA and protein levels in muscle during aging [106].

Irisin has pronounced protective effects on muscle tissue. Studies on C2C12 muscle cell cultures showed that irisin effectively prevents dexamethasone-induced myotube atrophy by suppressing the dephosphorylation of the transcription factor FoxO3α and reducing the expression of muscle-specific ubiquitin ligases Atrogin-1 and MuRF-1, which are responsible for the degradation of muscle proteins. Furthermore, irisin stimulates IGF-1-dependent signaling cascades, leading to enhanced protein synthesis and suppression of proteolysis in skeletal muscle, which is particularly important for maintaining muscle mass under conditions of age-related decline in anabolic signals [108]. One key mechanism of irisin action is its influence on mitochondrial homeostasis: it stimulates mitochondrial biogenesis via PGC-1α activation, improves mitochondrial dynamics, activates mitophagy to remove damaged mitochondria, and reduces oxidative stress, which is critical for preventing age-related mitochondrial dysfunction—a key mechanism in the development of sarcopenia [109].

Clinical studies demonstrate significant associations between circulating irisin levels and measures of muscle mass and function. A large cross-sectional study identified a positive correlation between serum irisin levels and appendicular muscle mass, handgrip strength, and overall physical function; individuals with established sarcopenia had significantly lower irisin levels, and low irisin levels were associated with an increased risk of sarcopenia [110]. A study of postmenopausal women showed that decreased irisin levels were associated with a reduced cross-sectional area of the quadriceps muscle, a higher prevalence of sarcopenia, and a 95% increased risk of sarcopenia for each 1.0 ng/mL decrease in irisin [111]. In patients with malnutrition, higher irisin levels were noted in those without asthenia, indirectly confirming the protective role of this myokine [112]. However, not all studies confirm this link: some work shows that irisin levels may be independent of the presence of sarcopenia in older adults [113], which may be related to differences in sarcopenia definition methods, population characteristics, and analytical approaches.

Physical exercise is an effective stimulus for increasing irisin levels in older adults. Studies show that exercise leads to increased basal irisin levels, improved muscle mass and strength indicators [114], and reduced markers of sarcopenia [115]. Studies in animal models confirm that aerobic exercise mediates beneficial effects on sarcopenia via Fndc5/irisin activation, improving muscle mass, strength, and metabolic parameters [116].

However, randomized clinical trials of irisin administration in humans are still lacking. Therefore, its potential is currently considered mainly in three directions: as a target for stimulating endogenous production through physical activity, as a component of diagnostic panels, and as a factor for stratifying sarcopenia risk. The roles and mechanisms of action of the key neurotrophic factors and myokines discussed in this section are summarized in Table 1.

The interactions among neurotrophic factors form a complex regulatory network. The collective impact of these factors on the presynaptic nerve terminal, postsynaptic muscle membrane, and satellite cells is summarized in Figure 3. An imbalance within this network, as occurs with aging, underlies the disintegration of the neuromuscular apparatus in sarcopenia.

### 3.3. Sarcopenia as a Component of the Neurodegenerative Disease Continuum: Shared Pathogenic Mechanisms with Alzheimer’s and Parkinson’s Diseases

#### 3.3.1. Alzheimer’s Disease

Sarcopenia and Alzheimer’s disease (AD) are age-associated conditions characterized by progressive impairment of muscle and nervous system function, respectively. Accumulating evidence points not only to an epidemiological link but also to shared pathophysiological mechanisms [117,118]. Of particular interest are the neurogenic mechanisms underlying sarcopenia development in neurodegenerative diseases, which opens new perspectives for understanding the interplay between the central nervous system and peripheral tissues during aging. Neurogenic sarcopenia is characterized by the loss of muscle mass and strength due to impaired muscle innervation, motor neuron dysfunction, and degradation of neuromuscular junctions (NMJs) [119,120]. In Alzheimer’s disease, such changes can arise as a consequence of central neurodegeneration or from the direct effects of pathological proteins on peripheral neuromuscular structures [121].

A key aspect is muscle denervation and NMJ dysfunction. Aging and neurodegenerative diseases are associated with progressive degradation of these structures, leading to functional muscle denervation [122]. Studies on a transgenic Alzheimer’s disease model (3xTgAD mice) demonstrated significant alterations in NMJ structure, including fragmentation and partial denervation, accompanied by reduced muscle contraction strength and accumulation of amyloid precursor protein (APP) and β-amyloid in muscle tissue [123].

Impaired nuclear membrane integrity and increased nuclear permeability, leading to disrupted nucleocytoplasmic transport, accumulation of toxic proteins in the nucleus, and ultimately neuronal death, represent a key mechanism of age-related motor neuron loss. Notably, physical activity can partially prevent these changes, indicating the modifiability of this mechanism [124] and opening avenues for therapeutic approaches aimed at sustaining motor neuron integrity.

Neuroinflammation plays a significant role in both sarcopenia and Alzheimer’s disease. Chronic low-grade inflammation, characteristic of aging (inflammaging), promotes the activation of catabolic pathways in muscle and neurotoxic processes in the central nervous system via pro-inflammatory cytokines such as TNF-α, IL-1, and IL-6 [125]. In Alzheimer’s disease, elevated pro-inflammatory mediators exert systemic effects on peripheral tissues, including skeletal muscle [126], creating a vicious cycle where central neuroinflammation exacerbates peripheral manifestations of sarcopenia.

Mitochondrial impairments represent a common pathogenic mechanism for sarcopenia and Alzheimer’s disease [127], including disruptions in mitochondrial dynamics, reduced biogenesis, defective mitophagy, and accumulation of mitochondrial DNA damage. In skeletal muscle, mitochondrial dysfunction leads to decreased energy provision for muscle contractions and activation of apoptotic pathways [128], whereas in the nervous system, impaired mitochondrial metabolism contributes to the accumulation of reactive oxygen species and disruption of neuronal energy homeostasis [129]. Furthermore, mitochondria play a critical role in maintaining NMJ function by providing energy for synaptic transmission and supporting the structural integrity of pre- and postsynaptic elements [122]. Age-related mitochondrial dysfunction thus contributes to NMJ destabilization and denervation.

#### 3.3.2. Parkinson’s Disease

Parallel to the mechanisms linking sarcopenia and Alzheimer’s disease, similar pathogenic interrelationships are observed in another neurodegenerative disorder—Parkinson’s disease (PD). Contemporary research demonstrates that PD and sarcopenia share common neurogenic developmental mechanisms based on fundamental molecular pathways, despite their distinct clinical manifestations [130,131]. Central to this relationship is α-synuclein dysfunction—a protein whose pathological aggregation is characteristic of PD but also significantly impacts peripheral neuromuscular structures.

Studies on mThy1-hSNCA transgenic mice showed that overexpression of human α-synuclein leads to its localization in motor neuron axons and NMJs, where it forms pathological aggregates. These aggregates directly disrupt NMJ structure and function, causing progressive denervation of muscle fibers. α-synuclein aggregation increases the number of abnormal mitochondria in intramuscular axons and NMJs by more than 60%, significantly impairing acetylcholine release from presynaptic vesicles [130]. This leads to a functional uncoupling of nerve and muscle, initiating a cascade of atrophic processes.

Mitochondrial dysfunction is a shared pathogenic mechanism of sarcopenia and PD. α-synuclein directly interacts with mitochondria, impairing the function of outer membrane transporters, particularly TOM40 (Translocase of Outer Membrane 40) [132], leading to disrupted protein import into mitochondria, reduced respiratory activity, and mitochondrial membrane depolarization. In muscle tissue from sarcopenic individuals, a significant decrease in the expression of genes and proteins involved in oxidative phosphorylation and mitochondrial biogenesis is observed, correlating with loss of muscle mass and function [127,133]. Oxidative stress and inflammation create a self-sustaining pathological cycle that enhances both α-synuclein aggregation and muscle tissue damage, with mitochondrial dysfunction leading to an approximately 220% increase in ROS production, significantly amplifying oxidative stress [130].

α-synuclein substantially alters the metabolic profile of satellite cells (MuSCs), the primary stem cells of skeletal muscle. Overexpression of α-synuclein reduces the basal respiratory capacity of MuSCs by approximately 170% and their glycolytic capacity by 150% [130]. These metabolic impairments significantly hinder the ability of MuSCs to migrate and fuse during muscle regeneration, reducing these processes by 60% and 40%, respectively. The altered metabolism of MuSCs disrupts normal muscle repair processes after injury, which is particularly critical in the context of age-related changes and neurodegenerative processes, where the need for effective muscle regeneration increases while the capacity for it declines.

Understanding the common mechanisms of sarcopenia and PD opens new therapeutic opportunities. The use of mitochondria-targeted antioxidants, such as Mito-TEMPO, shows promising results in restoring muscle regenerative capacity after injury in experimental models [130], highlighting the potential of targeted mitochondrial therapy for both conditions. The commonality of pathogenic mechanisms suggests the possibility of developing comprehensive therapeutic strategies aimed simultaneously at neuroprotection and preservation of muscle mass and function. Such an approach could include modulation of α-synuclein, improvement in mitochondrial function, reduction in oxidative stress, and support of regenerative processes.

For a comparative overview of the key pathogenic mechanisms shared by sarcopenia, Alzheimer’s disease, and Parkinson’s disease, see Table 2.

### 3.4. Summary: The Neuromuscular Junction as a Critical Link in the Pathogenesis of Muscle Failure

In summary, dysregulation of the neuromuscular axis is a central element in the pathogenesis of sarcopenia. As demonstrated, aging and sarcopenia involve progressive loss and functional reorganization of alpha-motoneurons, predominantly large ones innervating fast type II muscle fibers. A key pathogenic event is age-related denervation of muscle fibers, initiated by destabilization of the neuromuscular junction (NMJ). The presynaptically expressed motoneuron protein agrin (Agrn) plays a critical role in maintaining NMJ stability. By binding to the Lrp4/MuSK receptor complex on the postsynaptic membrane, agrin triggers a cascade of processes leading to the clustering of acetylcholine receptors (AChRs). Age-related reduction in agrin expression or impaired secretion by motoneurons is one of the initial molecular events leading to fragmentation of the postsynaptic apparatus and impaired synaptic transmission.

Alongside this, dysfunction of terminal Schwann cells, which normally provide structural and trophic support to the synapse, contributes significantly to NMJ degeneration. Age-related electrophysiological impairments, including reduced nerve conduction velocity and altered membrane excitability, are direct consequences of the described structural defects. Compensatory mechanisms, primarily collateral sprouting of surviving axons, although aimed at reinnervating denervated fibers and increasing motor unit size, become inadequate with age and cannot fully prevent the development of denervation atrophy and the shift in muscle fiber type distribution towards slow oxidative fibers.

The molecular basis of neurogenic sarcopenia includes dysregulation of neurotrophic factors, impaired proteostasis in motoneurons, and oxidative stress. Importantly, the identified pathological changes—loss of motoneurons, synaptic dysfunction, accumulation of pathological proteins—place sarcopenia within a common continuum with neurodegenerative diseases such as Alzheimer’s and Parkinson’s, suggesting shared molecular pathways.

Thus, current data convincingly demonstrate the neurogenic origin of sarcopenia. However, despite progress in understanding the structural and functional changes, a comprehensive picture of the molecular and genetic mechanisms initiating this process remains incomplete. This underscores the necessity of applying new analytical methods, including high-throughput sequencing technologies.

## 4. Genomic Landscapes Unveiled by NGS: Insights into Neurogenic Sarcopenia

The significant reduction in the cost of Next-Generation Sequencing (NGS) has revolutionized the study of complex polygenic diseases, providing unprecedented resolution for identifying coding and regulatory variants, structural variations (SVs), and copy number variations (CNVs). In the context of age-associated pathologies, these methods have revealed key genetic determinants of neurodegenerative diseases [134,135] and hereditary myopathies [136,137], creating a foundation for understanding the contribution of the neurogenic component to the development of sarcopenia.

Over the past five years, significant efforts have been made to identify the genomic basis of sarcopenia using various sequencing and association analysis approaches. However, the literature shows low reproducibility of results, explained by several factors: insufficient sample sizes, differences in study design, ethnic characteristics of populations, analysis of pre-selected candidate genes, and variability in patient clinical phenotypes. To minimize the risk of misinterpretation of the genomic factors of sarcopenia, this section exclusively reviews the results of Whole-Exome Sequencing (WES), Whole-Genome Sequencing (WGS), and Genome-Wide Association Studies (GWAS), which represent the most robust methodological approaches.

WES is a method focused on analyzing the protein-coding regions of the genome, which constitute about 1–2% of its total size [138], yet contain up to 85% of known disease-associated variants [139]. This method is based on the hybrid capture enrichment of exonic sequences followed by deep sequencing (coverage 95–160×) [140]. Its key advantages include cost-effectiveness (costing 10–20% of WGS), high depth of coverage ensuring reliable detection of even heterozygous and mosaic variants, simplified interpretation due to the focus on functionally significant regions, and good reproducibility thanks to standardized protocols. However, WES has significant limitations: it does not cover non-coding regions such as promoters, enhancers, and non-coding RNAs; it is characterized by uneven coverage, especially in GC-rich and repetitive regions; it has low sensitivity for detecting structural variations; and it presents interpretation challenges for genes with unknown function [141].

In contrast to WES, GWAS technology analyzes associations between phenotypic traits and genetic variants across the entire genome using pre-defined SNP microarrays [142]. This method provides genome-wide coverage without a priori hypotheses [143], allows for the analysis of large sample sizes, features standardized methodology, and enables the study of population-specific associations. Among the limitations of GWAS are the analysis of only pre-selected markers, the omission of rare variants, the problem of multiple testing, stringent significance thresholds, the explanation of only a small fraction of the heritability of complex traits, and strong dependence on the population composition of the sample [144].

The most comprehensive approach is WGS, which provides uniform coverage and sequences the entire genome without the bias associated with hybrid capture enrichment. WGS allows for the detection of all variant types—single-nucleotide variants, indels, structural rearrangements, and copy number variations—in both coding and regulatory regions of the genome. Its main advantages include complete coverage, the ability to analyze structural variations, and the detection of rare variants. However, WGS is significantly more expensive, requires substantial computational resources, creates challenges in interpreting the vast number of variants (especially in non-coding regions), and standard coverage (30×) may be insufficient for reliably detecting heterozygous variants in complex loci [145].

### 4.1. Genetic Determinants of Sarcopenia Development and Progression

The only review-period study employing WES was conducted on a sample of 101 patients from the Han Chinese population. This work analyzed the association of polymorphisms with whole lean body mass (WLBM)—a quantitative trait predictive of sarcopenia development. Although no variants reached genome-wide significance (GWS, *p* < 1.5 × 10^−7^), the strongest association was shown for the rs740681 polymorphism in the FZR1 gene (*p* = 1.66 × 10^−6^). Notably, four variants that did not reach GWS in the Chinese cohort were successfully replicated in the large-scale UK Biobank sample (n = 217,822). The functional significance of the FZR1 gene (Fizzy and cell division cycle 20 related 1), which encodes a substrate-specific adapter for the E3 ubiquitin ligase complex APC/C, lies in activating the degradation of cell cycle regulators. This suggests its role in controlling the terminal differentiation of muscle cells, consistent with the known importance of ubiquitin-dependent proteolysis in muscle tissue regulation. Another gene, SOAT2 (Sterol O-acyltransferase 2), is involved in lipid metabolism, and its dysregulation may affect muscle energy homeostasis [20].

In contrast to the WES approach, most research on the genetic architecture of sarcopenia and related phenotypes is based on GWAS of common variants. The first large GWAS of whole lean body mass, conducted in 2020, identified a significant locus at 3p27.1 (lead SNP rs3732593, *p* = 7.19 × 10^−8^) in the Framingham Heart Study cohort (n = 6004). However, replication in an independent sample showed only nominal significance (*p* = 0.04), potentially explained by the “winner’s curse” effect [146]. Bioinformatic analysis indicated that this locus regulates enhancer activity in skeletal muscle myoblasts and highlighted three candidate genes: MCF2L2 (involved in RhoA signaling, critical for actin reorganization and the cell cycle), B3GNT5 (involved in glycoprotein biosynthesis), and ATP11B (a P-type ATPase), whose functional link to muscle requires further study [147].

Parallel research has focused directly on identifying genetic predictors of clinically diagnosed sarcopenia. One such study in an elderly Taiwanese population identified 12 SNPs associated with key sarcopenia indices. Among the most significant were rs10282247 (gene OSBPL3, cholesterol binding) and rs7022373 (gene ACER2, apoptosis). An important result was the development of a polygenic risk score, where the presence of ≥4 risk alleles was associated with an exceptionally high disease risk (OR = 630.6), confirming its polygenic nature [148].

Further insight into genetic mechanisms was achieved in a large-scale meta-analysis of GWAS from Korean cohorts (n = 6961). This study not only identified new SNPs for lean body mass (LBM)—rs1187118 and rs3768582—and for appendicular skeletal muscle mass (ASM)—rs6772958, but also used eQTL analysis to identify associated genes (RPS10, NUDT3, NCF2, SMG7, ARPC5) demonstrating differential expression in skeletal muscle. Of particular interest is the enrichment of the mRNA destabilization pathway, pointing to impaired post-transcriptional regulation as a novel potential mechanism in sarcopenia pathogenesis, alongside dysregulation of lipid and energy metabolism [149].

Given the lack of effective pharmacotherapy for sarcopenia, studies aimed at identifying new therapeutic targets are especially valuable. A revolutionary approach was demonstrated by a study using Mendelian randomization to analyze approximately 16,000 drug-targeted genes. This identified genes negatively (BORCS7, UQCC1) and positively (PM20D1, NUCKS1) regulating muscle mass and function [150]. Additional analysis integrating GWAS/eQTL/pQTL data highlighted genes involved in inflammation (HP, HLA-DRA), cell signaling (MAP3K3, AURKA), extracellular matrix remodeling (COL15A1), and muscle regeneration (MFGE8) as promising therapeutic targets [151].

The problem of drug-induced sarcopenia is also beginning to receive genetic validation. An analysis of the FAERS database, followed by Mendelian randomization, identified several drugs associated with an increased risk of sarcopenia, including levofloxacin, pregabalin, statins, as well as metformin and NSAIDs. The genetic targets of these drugs include genes ETFDH, PRKAB1 (for metformin), and *PTGS1/PTGS2* (for aspirin/acetaminophen), providing important guidance for safer drug prescribing in elderly patients [152].

Contemporary research actively uses integrative approaches to study sarcopenia. Multi-trait analysis (MTAG) of walking speed and grip strength in the UK Biobank identified 20 novel loci and, using TWAS, pinpointed key genes PPP1R3A (associated with grip strength, encoding a regulatory subunit of protein phosphatase-1 involved in glycogen synthesis in skeletal muscle) and ZBTB38 (associated with walking speed, playing a critical role in initiating the myogenic program). Co-expression network and protein–protein interaction analyses further highlighted 11 genes, including ATP2A1 (muscle calcium homeostasis) [153]. The largest GWAS of muscle weakness to date (n = 256,523) confirmed the role of immune mechanisms, discovering a lead signal in the HLA-DQA1 locus, and found only partial genetic overlap with quantitative strength measures, indicating the uniqueness of the clinical phenotype. The second-most significant locus included the GDF5 gene (*p* = 4 × 10^−13^), encoding a TGF-β family protein with a key role in bone and joint development, previously associated with osteoarthritis and height [21].

Novel NGS technologies like WGS enable the detection of the contribution of rare variants. A 2025 study on a cohort including 10,729 samples identified seven novel loci associated with muscle mass, including sex-specific factors (LINC01661/PRMT6 in women, RCC2P8/COL25A1 in men), as well as rare variants in genes (DMAC1, ENSG00000273183). Functional validation in model organisms revealed potential mechanisms of action for the EMP2 and SSUH2 genes. Knockdown of EMP2 in *Drosophila* led to reduced motor activity without morphological muscle changes, while mutation of SSUH2 in Danio rerio stimulated muscle fiber hypertrophy without improving functional performance [154]. A parallel study in a Japanese population (n = 129) confirmed the significance of rare variants, identifying five genes (including SLC41A3, whose deletion disrupts the expression of myogenin (MYOG)—a key regulator of myogenic differentiation), and discovered a protective effect of the HLA-DPB1*02:01 allele, underscoring the role of immune regulation in sarcopenia. Notably, the HLA-DPB1*02:01 allele demonstrated a protective effect, particularly pronounced in Northeast Asian populations [155].

The final step in integrating this data is the development of polygenic risk models. A study in a Korean population (1368 cases vs. 15,472 controls) developed such a model based on five SNPs (FADS2_rs97384, MYO10_rs31574, KCNQ5_rs6453647, DOCK5_rs11135857, and LRP1B_rs74659977), which showed high predictive ability (OR = 1.977). A key finding was the demonstrative interaction of polygenic risk with age, metabolic syndrome, and grip strength, while regular physical activity was the only lifestyle factor capable of mitigating genetic predisposition [156]. This echoes the results of a pleiotropy analysis in the UK Biobank, which identified 78 SNPs associated simultaneously with three components of sarcopenia, as well as with obesity and type 2 diabetes. Risk alleles were shown to correlate with adverse phenotypes and behaviors, while protective alleles correlated with a healthy lifestyle. Importantly, strength training modulates the expression of many of these genes, explaining their therapeutic effect [157].

In conclusion, methods such as GWAS, WES, and WGS have significantly contributed to identifying pathogenic and protective factors for sarcopenia. However, current research faces fundamental limitations, including difficulties in interpreting variants in non-coding genomic regions and low statistical power for detecting associations of rare variants. Overcoming these limitations in the future will likely involve the widespread adoption of artificial intelligence and machine learning technologies for the integration and analysis of multidimensional genomic data. A summary of the key genetic studies on sarcopenia, including their design, sample size, and implicated loci, is provided in Table 3.

### 4.2. Epigenetic Regulation in Sarcopenia: The Role of DNA Methylation and Post-Translational Histone Modifications

Epigenetic mechanisms represent key regulators that integrate genetic background with environmental influences and age-related changes to modulate transcriptional programs relevant to neurodegeneration and muscle pathology. Epigenetic regulation includes DNA modifications (methylation), post-translational histone modifications, and alterations in chromatin accessibility, which are recurrently disrupted in Alzheimer’s disease, Parkinson’s disease, and various muscle disorders. These epigenetic changes affect the activity of promoters and enhancers, chromatin compaction, and the recruitment of transcription factors, thereby modulating neuronal survival, synaptic function, inflammation, and muscle fiber identity [158,159]. The ChIP-seq method, which analyzes DNA–protein interactions, maps transcription factor binding and histone modifications across the genome, enabling the identification of active and repressive chromatin states and linking chromatin marks to transcriptional dysregulation in diseases. ChIP-seq identifies disease-associated shifts in activating (e.g., H3K9ac) or repressive (e.g., H3K27me3) histone marks at promoters/enhancers [160]. The ATAC-seq method profiles open chromatin and nucleosome positioning to identify putative enhancers, promoters, and cell-specific regulatory landscapes. For instance, chromatin accessibility profiling in iPSC-derived motor neurons from ALS patients revealed disease-specific changes in regulatory regions [161]. Bisulfite sequencing maps cytosine methylation genome-wide at single-nucleotide resolution, revealing DNA methylation patterns often disrupted in neurodegenerative and muscle diseases. For example, methylome analysis identified hypomethylation of Ambra1, which exacerbates dopaminergic neuron damage in Parkinson’s disease mouse models [162].

Despite rapid advances in epigenomics, the application of ChIP-seq for analyzing histone modifications in the context of sarcopenia remains extremely limited. The only review-period study using this approach revealed the role of β-hydroxybutyrate (β-HB) and the post-translational histone modification it induces—β-hydroxybutyrylation (Kbhb)—in the pathogenesis of skeletal muscle aging. Integrated RNA-seq and ChIP-seq analysis demonstrated that exogenous β-HB increases Kbhb histone modifications (H3K9bhb and H3K27bhb) in the promoter regions of mitochondrial biogenesis genes. The activation of genes involved in oxidative phosphorylation, ATP metabolism, and aerobic respiration was accompanied by the restoration of mitochondrial function and organellar structural integrity, which is critical for energizing neuromuscular transmission. Inhibition of Kbhb using the p300 inhibitor A485 (in vitro) and *cbp-1* knockdown (in *C. elegans*) completely abolished the protective effect of β-HB on myofibrils and motor function. These results position Kbhb not only as a biomarker but also as a functional mediator of anti-sarcopenic action. Although the study was performed on model organisms (*C. elegans*, mice), limiting extrapolation to humans, it provides unique evidence for the role of this modification in protecting against sarcopenia and sets new research directions [163].

Although direct ATAC-seq studies in neurogenic sarcopenia are lacking, data from agricultural livestock reveal evolutionarily conserved mechanisms of epigenetic regulation. A study in cattle [164], integrating RNA-seq and ATAC-seq, identified 4564 muscle-specific open chromatin regions, predominantly localized in promoters of genes associated with muscle structure development (GO:0061061), myocyte differentiation (GO:0055001), and neuromuscular signaling (GO:0003012). Critically significant was the enrichment of these regions in cAMP, cGMP-PKG, and MAPK signaling pathways, which regulate key aspects of neuromuscular homeostasis: motor neuron apoptosis (MAPK), calcium cycling in the sarcoplasmic reticulum (cGMP-PKG), and neurotrophin expression (cAMP). Integrative analysis highlighted 54 regulator genes coordinated by the transcription factor MEF2C, including genes for the contractile apparatus (ACTN2, MYOM3), epigenetic modifiers (DOT1L), and regulators of proteolysis (UBE2G1). Notably, MEF2C directly activates the expression of HRC (Histidine Rich Calcium Binding Protein), which controls calcium homeostasis, fundamentally important for neuromuscular junction function.

These findings are complemented by studies on age-related dysfunction of porcine satellite cells [165]. ATAC-seq revealed a progressive reduction in chromatin accessibility upon passaging (F7 vs. F3), characterized by 767 loss-DARs (differentially accessible regions) primarily in intergenic regions. These regions contained binding motifs for Atf4—a factor associated with muscle atrophy. Despite a weak global correlation between chromatin accessibility and gene expression (r < 0.3), critical changes were detected: decreased expression of FGD5 (a regulator of PI3K signaling) was accompanied by the closing of its promoter, and suppression of SLIT3 (necessary for myogenic differentiation) correlated with loss of muscle mass. A key pathogenic mechanism was the activation of the p53-p21-RB pathway, leading to cell cycle arrest in the G1 phase.

In contrast to the dynamic changes in chromatin accessibility revealed by ATAC-seq and ChIP-seq, alterations in DNA methylation patterns represent stable epigenetic modifications that accumulate systemically with aging, forming the molecular basis of sarcopenia. A study based on whole blood from 50 sarcopenia patients [166] identified hypomethylation of a specific region in the FGF2 gene (β = −0.42, *p* < 0.001), the level of which correlated with disease severity and demonstrated diagnostic value. These data are consistent with a Korean study [167], where integration of the methylome with clinical muscle mass indices revealed sex-specific differentially methylated regions (DMRs) in the GAB2, JPH3, and *HLA-DQB1* genes, enriched in Rap1 signaling and glutamatergic synapse pathways—key for neuromuscular transmission. Furthermore, a population-based study conducted in the Xinjiang Uygur Autonomous Region [168] demonstrated hypomethylation of CpG sites in the *TWEAK/Fn14* promoter (CpG8/12/13/20/21) in sarcopenia patients, which correlated with increased plasma levels of TWEAK (*p* = 0.007) and TNF-α (*p* < 0.001), confirming the role of epigenetically mediated neuroinflammation in sarcopenia pathogenesis.

At the skeletal muscle level, a large-scale analysis of m. vastus lateralis biopsies (Hertfordshire Sarcopenia Study) [169] identified numerous differentially methylated CpG sites and DMRs enriched in genes for myotube fusion, oxidative phosphorylation, and calcium channels. Functional validation confirmed the causal role of these changes. One of the most interesting findings was the confirmation that inhibition of the epigenetic regulator EZH2 in primary myoblasts impaired differentiation and mitochondrial biogenesis. These data are supported by a meta-analysis of 908 muscle methylomes [170], which revealed hypermethylation of Polycomb target genes and hypomethylation of enhancers. The study by Turner et al. (2020) identified global DNA hypermethylation in the skeletal muscle of older adults (83 ± 4 years) compared to young individuals (27 ± 4.4 years), particularly in key oncogenic signaling pathways such as focal adhesion, MAPK, PI3K-Akt-mTOR, p53, Jak-STAT, TGF-beta, and Notch. Key targets of age-related changes were genes of the HOX family, namely *HOXD10, HOXD9, HOXD8, HOXA3, HOXC9, HOXB1, HOXB3, HOXC-AS2*, and HOXC10, which showed differential methylation [171]. These epigenetic changes correlated with impaired morphological differentiation of muscle cells and reduced expression of the key myogenic factors MyoD and Myogenin [171].

Gender differences in epigenetic regulation are of particular interest. A study in women identified global blood hypomethylation in sarcopenia (Δβ = −0.004, *p* < 0.05) with hypermethylation at TSS200 (HSPB1, CNKSR3) and hypomethylation at 3′UTRs, affecting cytoskeleton regulation pathways [172]. In a Korean male cohort [173], machine learning identified diagnostically significant CpG sites, and a calculated Methylation Risk Score (MRS) correlated with age and decreased muscle mass/strength. Contradictions in global patterns are illustrated by a Polish study [174], which found a paradoxical increase in global blood methylation in sarcopenia, likely reflecting methodological differences and the influence of confounding factors.

Synthesizing current data allows for the proposal of a unified model of epigenetic dysregulation in sarcopenia, integrating four interconnected pathogenic axes: pro-inflammatory, neurotrophic, myogenic, and calcium-dependent. The identified epigenetic biomarkers form the basis for developing multi-omics diagnostic panels, while therapeutic prospects encompass both pharmacological modulation of epigenetic targets and non-pharmacological interventions.

## 5. Transcriptomic Signatures from NGS: Dysregulated Pathways in Neuromuscular Crosstalk

### 5.1. Systemic Analysis of Muscle Tissue Transcriptome: Dysregulation of Key Signaling Pathways and Biomarker Identification

High-throughput transcriptomic analysis of skeletal muscle is a powerful tool for identifying key molecular pathways involved in the pathogenesis of sarcopenia. Studies reveal complex changes affecting a wide spectrum of biological processes. Specifically, analysis of sarcopenia development stages shows that the manifest stage of the disease is associated with dysregulation of Gonadotropin-Releasing Hormone (GnRH), Neurotrophin, Rap1, Ras, and p53 signaling pathways, indicating the involvement of endocrine, neurotrophic, and apoptotic mechanisms in pathogenesis. At the stage of isolated low muscle mass, activation of Vascular Endothelial Growth Factor (VEGF) signaling, B- and T-cell receptor signaling, and ErbB receptor cascades is observed, suggesting compensatory activation of angiogenic and immunoregulatory mechanisms. Progression to the stage of combined low muscle mass and function shows reduced activity of the B-cell receptor, apoptosis, HIF-1α signaling, and adaptive immune response pathways, indicating profound immune and metabolic dysregulation. In the same analysis, TTC39DP, SLURP1, LCE1C, PTCD2P1, and OR7E109P were identified as potential biomarkers. Notably, the expression of SLURP1 and LCE1C shows ethnic specificity, reaching maximum values in patients of Chinese origin compared to European and Afro-Caribbean populations. Regulatory analysis of the most activated genes revealed a key role for GATA family transcription factors as master regulators controlling the expression of predicted genes [175].

A growing body of experimental evidence underscores the critical role of immune dysregulation in sarcopenia pathogenesis. Integrative co-expression analysis demonstrates a close relationship between transcriptomic changes and immune homeostasis disturbances in this disease. Comprehensive analysis of immune-associated transcriptional regulatory networks identified a potential regulatory axis, PAX5-SERPINA5-PI3K/Akt, in which B-lymphocytes are presumed to play a key role. This discovery highlights the central importance of immune dysregulation not only at the systemic level but also within specific cell-signaling cascades in muscle tissue. Based on the established transcriptomic profile, a screen for potential therapeutic agents was conducted, identifying the histone deacetylase inhibitor Trichostatin A as a promising compound for sarcopenia therapy [176].

Parallel to the identification of general transcriptomic patterns, there is an ongoing search for specific driver genes of sarcopenia using advanced computational methodologies. The use of Artificial Neural Networks (ANNi) to analyze four independent transcriptomic datasets identified the top 200 genes influencing aging processes or being affected by them. This analysis revealed CHAD, ZDBF2, USP54, and JAK2 as genes with the strongest interactions predictive of muscle aging, while SCFD1, KDM5D, EIF4A2, and NIPAL3 were identified as the main interactive genes associated with long-term physical exercise in older adults. Validation by quantitative real-time PCR (RT-qPCR) confirmed a statistically significant increase in the expression of USP54, CHAD, and ZDBF2 in aging muscles. Gene Ontology analysis predicted enrichment of pathways related to immune response, apoptosis, and bone tissue development in aging, while physical exercise in older adults was associated with extracellular matrix remodeling and proteolysis pathways. These driver genes represent novel targets for pharmacological and non-pharmacological interventions [177].

One of the central pathophysiological processes in sarcopenia is mitochondrial dysfunction. Transcriptomic studies consistently show significant suppression of pathways related to “mitochondrial ATP synthesis coupled to electron transport” and “NADH dehydrogenase complex assembly.” This leads to reduced energy production, a common pathological feature in individuals with sarcopenia. Notably, current data indicate that impaired function of muscle stem cells (satellite cells), rather than the muscle fibers themselves, may be a key factor disrupting the skeletal muscle repair function and contributing to disease development. Pharmacoinformatic analysis aimed at finding drugs capable of counteracting these transcriptomic changes pointed to the potential of histone deacetylase inhibitors. One representative of this group, Vorinostat, enhances mitochondrial function and stimulates myoblast differentiation in vitro, opening avenues for drug repurposing [178].

Mitochondrial dysfunction is closely interrelated with impairments at the neuromuscular junction (NMJ). Transcriptomic analysis reveals denervation in aging skeletal muscles, with the loss of satellite cell contribution being a driving force behind age-related NMJ degeneration. Satellite cells serve as a source of postsynaptic myonuclei, and their ability to replenish the postsynaptic region is impaired with age. Accelerated loss of satellite cells exacerbates the reduction in postsynaptic myonuclei, NMJ integrity, and muscle fiber size. Conversely, increasing the satellite cell pool through Spry1 protein overexpression specifically in satellite cells is associated with improved NMJ integrity and force generation in aging muscles. This underscores that the loss of the regenerative potential of muscle fibers provided by satellite cells is a significant mediator of age-related NMJ degeneration, rather than merely a consequence of denervation [179].

The role of denervation as a model of muscle atrophy has been studied in detail. Fourteen days after tibialis muscle denervation, extensive transcriptome remodeling is observed: genes related to oxidative phosphorylation, the tricarboxylic acid cycle, glycolysis/gluconeogenesis, and angiogenesis are suppressed, indicating profound inhibition of energy metabolism. Simultaneously, pathways of ubiquitin-mediated proteolysis, apoptosis, and aging are activated. This model confirms that impairments in energy metabolism and activation of proteolytic systems are universal mechanisms shared by various types of muscle atrophy, including sarcopenia [180].

The paradigm for targeted therapy in sarcopenia is undergoing significant transformation. Contrary to traditional views based on short-term hypertrophy models suggesting the need for mTORC1 activation, accumulating evidence indicates that efforts should focus on its suppression. It has been demonstrated that mTORC1 hyperactivity in skeletal muscles meets all criteria to be considered a key hallmark of sarcopenia. mTORC1 activity is increased in atrophied myofibrils of sarcopenic mice; sustained mTORC1 activation replicates age-related changes in neuromuscular junction (NMJ) stability and gene expression profiles; and long-term suppression of mTORC1 with rapamycin reliably slows age-related loss of skeletal muscle size and function. These data point to the role of mTORC1 hyperactivation as a hallmark of sarcopenia [181].

To delve deeper into mechanisms specific to the NMJ, a transcriptomic analysis of synaptic (SR) and non-synaptic (NSR) regions of the mouse diaphragm was conducted. Numerous genes differentially expressed between SR and NSR in newborns and adults were identified, along with significant changes in the SR transcriptome during development. The analysis predicted a key role for pathways such as “ion channel activity” and “neuroactive ligand–receptor interaction” for NMJ development, and pathways like “myelination” and “voltage-gated ion channel activity” for its maintenance. This study provides a valuable resource for understanding the molecular mechanisms underlying NMJ development and maintenance, the disruption of which is a cornerstone in sarcopenia pathogenesis [182].

Beyond general mechanisms, sexual dimorphism in the transcriptomic response to aging is of significant importance. Analysis of the human lateral gastrocnemius revealed distinct gene expression profiles in aging men and women. In women’s muscles, pathways of glucose catabolism, NAD metabolism, and fiber-type switching are impaired with age, while in men, pathways of replicative senescence, cytochrome C release, and muscle composition are predominantly affected. Unique candidate genes for each sex were identified (e.g., KIF20A, PIMREG in women and CENPK, CDKN2A in men), indicating the potential need for different therapeutic strategies to combat sarcopenia in men and women [183].

Caloric restriction (CR) is considered a promising therapeutic strategy for modulating sarcopenia. Transcriptomic analysis of rat muscles showed that CR suppresses 69.7% of genes upregulated with aging and restores the expression of 57.8% of downregulated genes. Furthermore, CR uniquely regulates many other genes. Key hub genes involved in both aging (e.g., Fgg, Fga) and the response to CR (e.g., Gc, Plg, Irf7, Alb, Apoa1) were identified. These data provide initial evidence for numerous targets for future therapeutic interventions aimed at mimicking the beneficial effects of CR [184].

Finally, an important direction is the development of models for screening and therapy. Exposing a human muscle-on-a-chip model to microgravity for 7 days reproduces key aspects of impaired myogenesis: a metabolic shift towards lipid metabolism, increased expression of pro-apoptotic genes (BOK, SGK1), suppression of Notch signaling, and increased levels of the GDF-15 protein—a marker of cellular senescence. The transcriptomic profile showed significant similarity to that of sarcopenia. This model was validated for drug screening: the addition of pro-regenerative drugs—IGF-1 and a 15-PGDH inhibitor—partially counteracted the negative effects of microgravity, confirming the utility of such platforms for discovering new therapeutic targets [185].

In conclusion, whole-tissue transcriptomic analysis demonstrates the existence of a wide diversity of pathogenic mechanisms in sarcopenia, including impaired immune regulation, mitochondrial dysfunction, neuromuscular junction degeneration, and profound metabolic reprogramming. As summarized in Figure 4, each of these core pathophysiological pathways presents a potential target for therapeutic intervention, with specific treatments—such as histone deacetylase inhibitors for immune dysregulation, compounds like Vorinostat for mitochondrial dysfunction, rapamycin for mTORC1 suppression, and strategies to enhance satellite cell function for NMJ stability—showing efficacy in preclinical models. Data integration using bioinformatic and machine learning methods allows for the identification of key driver genes and potential therapeutic intervention points, including the repurposing of existing drugs and the development of sex-specific approaches for treating this pathology.

### 5.2. Analysis of Cellular Heterogeneity by Single-Cell Sequencing: The Contribution of Cellular Populations to Sarcopenia Development

Traditional bulk RNA studies have provided valuable information on transcriptomic changes associated with sarcopenia, shedding light on key signaling pathways. However, this approach has a fundamental limitation—it provides averaged gene expression data for the entire tissue, masking cellular heterogeneity and the specific contribution of different cell populations to disease pathogenesis. Given that innervated skeletal muscle is a complex structure composed of multiple cell types, including muscle fibers, motoneurons, MuSCs, fibro-adipogenic progenitors (FAPs), immune cells, and endothelial cells [186,187], averaging signals from these heterogeneous populations can obscure critically important pathogenic mechanisms of sarcopenia. Advances in single-cell and single-nucleus RNA sequencing (sc/snRNA-seq) technologies have overcome this limitation, enabling a shift from studying averaged transcriptomic signatures to a detailed characterization of the cellular heterogeneity of skeletal muscle in sarcopenia.

A combined approach using both bulk and single-nucleus RNA-seq revealed gene expression changes occurring with age and the onset of frailty across all cell types of mature skeletal muscle. Specifically, aging muscle showed increased expression of genes such as MYH8 (embryonic myosin heavy chain) and PDK4 (pyruvate dehydrogenase kinase 4, which suppresses oxidative metabolism), alongside decreased expression of IGFN1. Furthermore, a small but significant population of nuclei expressing the cellular senescence marker CDKN1A (p21cip1) was identified, present only in samples from older individuals, directly implicating senescence in the pathogenesis of age-related frailty [23].

The creation of a comprehensive Human Muscle Ageing Cell Atlas (HMA) based on snRNA-seq revealed profound changes in the composition and functional state of all cellular populations. Aging muscle exhibits a loss of specific myonuclei types and the emergence of new subtypes, global activation of inflammatory and catabolic programs, disrupted expression of contractile protein genes, and transcriptome remodeling dependent on muscle fiber type. Specifically, myonuclei from type I fibers show metabolic reprogramming towards glycolysis, while myonuclei from type II fibers exhibit enhanced protein catabolism, explaining their greater susceptibility to atrophy. Beyond myonuclei, stromal cells are also significantly altered: the muscle stem cell (MuSC) pool shrinks and shows chronic activation of stress signaling, fibro/adipogenic progenitors (FAPs) switch from a pro-regenerative to a profibrotic profile, and immune cells enhance inflammatory programs, collectively creating a niche imbalance that promotes sarcopenia development [18].

Understanding the molecular basis of sarcopenia requires unraveling the fundamental transcriptional programs governing muscle development and maturation. A multi-omic snRNA-seq and ATAC-seq atlas of mouse muscle development revealed key gene regulatory networks. During muscle fiber formation, the Myogenin-Klf5-Tead4 complex synergistically activates muscle gene expression. Subsequently, during fiber maturation, the transcription factor Maf acts as a molecular switch, activating the mature gene program for fast-twitch fibers. Importantly, the expression of Maf and, consequently, fiber maturation were impaired in mutant mice lacking L-type voltage-gated calcium channels (Cav1.1), revealing a direct genetic link between neural activity, muscle contraction, and the transcriptional program of muscle fiber maturation [188]. These findings are highly relevant to sarcopenia, as age-related denervation and reduced neuromuscular transmission may disrupt such key transcriptional programs, contributing to the loss of muscle mass and function.

A key aspect revealed by single-cell sequencing is the state of cellular senescence within the muscle niche. A multi-omic study (snRNA-seq + snATAC-seq) provided the first map of senescence in mononuclear cells of aging human muscle. A Unified Senescence Score (USS) was developed, revealing significant heterogeneity both between different cell types (MuSCs, FAPs, endothelial, and smooth muscle cells) and within them. Analysis of SASP (senescence-associated secretory phenotype) factors showed their unique composition for each population, with a common core of inflammatory chemokines and cytokines such as CCL2, CCL3, CCL4, IL-6, and IL-1β. Transcriptomic and epigenomic analysis identified key transcription factors regulating senescence and SASP, dominated by members of the AP-1 family (ATF3, JUNB) acting alongside the known regulator NF-κB. Blocking one key SASP pathway (CCL3/4/5) using the antagonist maraviroc (MVC) in vivo led to significant improvement in muscle function and reduced senescence burden in old mice, confirming the central role of the senescent cell secretory phenotype in sarcopenia pathogenesis and opening avenues for therapeutic interventions [189].

The status and behavior of muscle stem cells (MuSCs) are of particular interest in the context of sarcopenia. It was discovered that the MuSC population performs a compensatory neuroreactive function. During aging and neuromuscular degeneration (in the *Sod1^−/−^* model), these cells are activated in response to synaptic changes. In young muscle post-injury, MuSCs can specifically engraft in a position proximal to the NMJ, likely a mechanism for supporting its integrity. However, this ability is lost in aging muscle, contributing to NMJ degeneration and sarcopenia progression [190]. Single-cell analysis of human MuSCs revealed their division into discrete subpopulations of ‘quiescent’ and ‘early-activated’ cells. Activated MuSCs display characteristic transcriptional profiles associated with aging, obesity, diabetes, and impaired muscle regeneration, as well as enrichment of the TWEAK-FN14 pathway, typical of muscle wasting conditions. In contrast, quiescent MuSCs express the marker EGFR [191]. It should be noted that certain discrepancies exist between studies: Rubenstein et al. [192] observed one MuSC population from dissociated whole-muscle samples, whereas Barrueta et al. identified 12 clusters from human MuSCs [193]. These discrepancies may be related to differences in cell isolation methods and bioinformatic analyses.

The role of immune cells, particularly macrophages, in creating a pro-inflammatory microenvironment in sarcopenia has also been detailed using snRNA-seq. Analysis of over 100,000 nuclei from muscles of young, old, and sarcopenic individuals identified 10 major cell types and 6 immune cell subclusters. Macrophages constitute the largest fraction of immune cells in the muscle microenvironment, and their composition and communication change substantially with age. Four subpopulations of resident macrophages playing different roles were identified, along with a skeletal muscle-specific marker for resident macrophages—LYVE1. Bulk analysis confirmed the enrichment of pathways associated with macrophage inflammation in sarcopenia, pointing to macrophages as a potential therapeutic target [194].

Beyond changes in myonuclei and stromal cells, single-cell technologies have revealed the important role of other resident populations. In a mouse model of sarcopenia (SAMP6), scRNA-seq of the tibialis muscle identified 14 cellular clusters. The dominant changes were found not in myocytes but in endothelial cells: a specific subpopulation of endothelial cells was discovered, characterized by a significant increase in the expression of interferon response genes and Guanylate-Binding Protein (GBP) family GTPases, such as Igtp and Gbp2. Functional validation in human cells showed that overexpression of Gbp2 impaired the ability of endothelial cells to form tubes, indicating impaired angiogenesis as a possible novel mechanism contributing to sarcopenia via the interferon-GBP signaling pathway [195].

Specific attention is given to transcriptomic remodeling at the neuromuscular junction. Applying snRNA-seq to synaptic and extrasynaptic myonuclei identified numerous NMJ-specific transcripts and showed that their expression is regulated not only by the agrin-Lrp4/MuSK signal but also by electrical activity and other trophic factors. Several new regulators of NMJ stability were characterized: overexpression of the transcription factor Etv4 was sufficient to activate ~50% of NMJ genes in extrasynaptic nuclei, and knockout of the Pdzrn4 gene, which encodes a protein interacting with MuSK in the Golgi apparatus, induced NMJ fragmentation. These data provide a rich resource for understanding the molecular basis of NMJ stability at the single-cell level, the disruption of which is a cornerstone of sarcopenia [196].

In summary, single-cell transcriptomic studies depict sarcopenia as a complex systemic disease of the muscle niche, involving coordinated changes in the state and communication of all resident cell populations—from myonuclei and stem cells to endothelium, immune, and stromal cells. MuSC dysfunction, the accumulation of senescent cells with a pro-inflammatory SASP, impaired neuromuscular transmission, and angiogenesis create a vicious cycle leading to progressive loss of muscle mass and function. The identification of key regulatory nodes, such as AP-1-dependent SASP transcription or interferon-GBP signaling in endothelium, opens new perspectives for targeted therapeutic strategies aimed at interrupting this cycle and preserving muscle homeostasis in old age. These complex, coordinated shifts in the cellular landscape of the aging neuromuscular axis are summarized schematically in Figure 5. Despite significant progress, discrepancies exist between studies, such as the reported number of MuSC subpopulations, potentially due to differences in cell isolation methods, sequencing platforms, and bioinformatic analyses. Furthermore, controversies remain regarding the contribution of different cell types to sarcopenia pathogenesis, warranting further investigation.

## 6. Translational Implications and Future Directions: From NGS Data to Clinical Applications

### 6.1. Contemporary Experimental Models: Platforms for Modeling Sarcopenia In Vitro and In Vivo

The development of relevant experimental models is a critical step in researching the molecular mechanisms of sarcopenia and screening potential therapeutic strategies. Contemporary methodological approaches encompass a wide spectrum of platforms—from in vitro cell cultures to complex in vivo models—each contributing uniquely to understanding disease pathogenesis.

Traditional two-dimensional (2D) cultures of immortalized cell lines, such as rat L6 [197] and mouse C2C12 [198] myoblasts, remain widely used tools due to their simplicity, reproducibility, and cost-effectiveness. These systems allow for the study of fundamental cellular processes and initial screening of pharmacological compounds. Their principal limitation is the inability to replicate the complex architecture and functional maturity of native muscle tissue, which reduces their predictive value for clinical outcomes. To mimic sarcopenia pathophysiology, various inductors are used in these models, including oxidative stress (H_2_O_2_), pro-inflammatory cytokines (TNF-α, IL-1β), dexamethasone, ionizing radiation, and sphingophospholipids (ceramide and palmitate) [199,200]. These agents enable the modeling of various aspects of muscle atrophy and the study of molecular mechanisms involved in disease development.

To enhance physiological relevance, human primary myoblasts isolated from muscle biopsies are increasingly used. These cells retain donor characteristics, including age-related features and genetic predispositions. For instance, primary myoblasts from insulin-resistant patients exhibit defects in insulin signaling [201]. Studies on cells from paraspinal muscles of patients (aged 25–77) demonstrate a clear age-dependent correlation: a decline in proliferative and differentiation potential correlates with increased levels of cell cycle inhibitors (p16INK4a, p21CIP1, p27KIP1) and inactivation of key myogenic transcription factors (MyoD, myogenin, MEF2c) [202]. A significant achievement has been the creation of three-dimensional bioartificial skeletal muscles based on human primary myoblasts cultured to replicative senescence [203]. Such models display key aging markers (SA-β-Gal, DNA damage), significant reduction in contractile function, impaired calcium signaling, and loss of regenerative potential after injury, replicating key aspects of sarcopenia.

A further development in this direction is the use of human induced pluripotent stem cells (iPSCs). Modern directed differentiation protocols allow for the generation of functional myogenic progenitor cells capable of forming three-dimensional contractile structures. A promising avenue is the creation of three-dimensional neuromuscular organoids (NMOs) based on human iPSCs [204]. These complex systems replicate the three-dimensional tissue architecture, cellular diversity, and functional connections between motor neurons and muscle fibers. A key advantage is the ability to use patient-specific iPSCs, enabling the modeling of genetic forms of neuromuscular diseases. For example, organoids derived from iPSCs of patients with a C9orf72 mutation replicate key pathophysiological features of ALS, including impaired skeletal muscle contractile function, denervation, and the accumulation of dipeptide repeats [205].

For a more realistic representation of muscle, co-culturing different cell lines to create three-dimensional bioengineered constructs resembling the extracellular matrix is employed [206]. Artificial muscle fibers based on collagen prolong the quiescence period of muscle stem cells, and optogenetically encoded C2C12 myoblasts allow testing the influence of various factors on muscle maturation, structure, and function upon optical stimulation [207]. Three-dimensional neuromuscular co-cultures show that innervation occurs four times faster and more efficiently compared to 2D models, and the functional integration of the adult acetylcholine receptor epsilon subunit occurs only under 3D culture conditions [208].

To further increase complexity and physiological relevance, “muscle-on-a-chip” systems are being developed. These devices integrate 3D cell cultures into microfluidic channels, allowing precise control of the microenvironment and enabling chronic exposure studies. Such platforms also facilitate the study of extreme environmental factors, such as microgravity. Experiments under spaceflight conditions revealed significant downregulation of key structural proteins (MYH1, ACTN3) and increased sensitivity of myotubes from older donors to the dysregulation of immune mediators [209].

Despite progress in in vitro methods, animal models remain indispensable for studying systemic aspects of sarcopenia. The similarity of aging processes in mice and humans has been demonstrated, making these models valuable for testing drug efficacy and therapeutic approaches [210]. C57BL/6J mice (older than 18 months) serve as a model of natural aging and allow investigation of the molecular mechanisms of sarcopenia, including oxidative stress, mitochondrial dysfunction, and protein synthesis/degradation systems. An important finding has been the identification of significant sexual dimorphism: male C57BL/6J mice exhibit more pronounced sarcopenia with marked functional decline, impaired contractility, reduction in type IIB fibers, and suppressed AMPK-autophagy, while these changes are less pronounced in aging females [211]. Senescence-accelerated mouse (SAM) strains provide a cost-effective alternative to natural aging models. SAMP strains are characterized by accelerated aging, with features similar to symptoms observed in elderly humans [212]. Specifically, the SAMP8 line demonstrates early and pronounced development of sarcopenia against a background of neurodegeneration, making it particularly relevant for studying the link between the nervous system and muscle atrophy. For example, SAMP8 mice exhibit symptoms of Alzheimer’s disease [213]. SAMP8 mice at 8 months of age are considered to be in the presarcopenia stage, while 10-month-old animals correspond to the sarcopenia stage [214]. These models are successfully used to investigate new sarcopenia treatments and assess their efficacy.

The use of genetically modified mice contributes to understanding specific mechanisms of sarcopenia. Studies on mice with double knockout of 4EBP1/2 revealed the role of muscle protein synthesis regulation in sarcopenia pathogenesis [215]. Growth hormone receptor knockout mice (GHR^−/−^) provide evidence for the role of the IGF-1 signaling pathway in regulating myostatin [216]. Opa1^−/−^ mice allow the study of the correlation between mitochondrial function and muscle atrophy [217], while superoxide dismutase-deficient mice (Sod1^−/−^) exhibit increased mitochondrial hydroperoxide production leading to muscle atrophy [218].

Various model organisms are used in sarcopenia research, including *Drosophila*, *Caenorhabditis elegans*, and *zebrafish*. The fruit fly *Drosophila* is an excellent model for aging research due to low maintenance costs and the possibility of genetic manipulation; however, the absence of MuSCs limits its applicability. *C. elegans* is used to model sarcomere assembly, maintenance, and regulation. They are isogenic and exhibit stochastic aging, similar to humans. *Zebrafish*, in turn, show aging comparable to humans and are suitable for studying age-related changes in the musculoskeletal system. The presence of satellite cells makes this model more similar to humans [199,219].

In conclusion, the future of sarcopenia modeling lies in the development of integrated approaches combining the advantages of various models. Priority directions include the creation of multi-organ systems, the development of standardized protocols for generating vascularized organoids, and the refinement of personalized models based on specific patient iPSCs. The improvement in neuromuscular interaction models is of particular importance, given the key role of impaired neural regulation in the pathogenesis of age-related muscle atrophy. Integrating data obtained using relevant experimental approaches will enable a holistic understanding of sarcopenia pathogenesis and the development of effective therapeutic strategies. An important addition to traditional methods will be in silico models using computational methods and databases to predict phenotype–genotype correlations and transcriptomic analysis. Although organoids modeling skeletal muscle and the neuromuscular apparatus are still under active development, they already demonstrate significant progress in replicating complex pathophysiological processes. Further development of these technologies will overcome the limitations of traditional model systems and facilitate a transition to a new level of understanding of sarcopenia mechanisms, opening possibilities for the development of effective personalized therapeutic strategies.

### 6.2. Biomarker Identification Strategies: Integration of Multi-Omics Data for Diagnosis and Treatment

High-throughput sequencing methods have enabled the identification of complex molecular-genetic signatures defining the pathogenic architecture of sarcopenia. The key challenge at the current stage is translating these discoveries into therapeutic strategies based on a deep understanding of the disease. The convergence of multi-omics data with clinical phenotyping will potentially redefine sarcopenia as a spectrum of molecular subtypes requiring personalized interventions. However, despite the significant volume of accumulated omics data for sarcopenia, their integration is hampered by the methodological heterogeneity of research studies. Key persistent problems include: variations in study designs and sequencing platforms, biological variability of muscle biopsies, non-standardized approaches to accounting for covariates and statistical analysis. Furthermore, an important limitation is the static nature of each omics approach individually, as each provides a snapshot of the molecular picture at a single point in time.

A promising direction is the development of minimally invasive biomarkers for early detection and patient stratification. For example, the C-terminal agrin fragment (CAF), whose level correlates with neuromuscular junction degradation, can be measured in serum [220,221]. Similarly, inflammatory transcriptional signatures (e.g., GDF15) are detectable both in muscle tissue and peripheral blood [222]. Implementing these biomarkers into clinical practice requires standardization of analytical methods and conducting longitudinal studies to validate their predictive value. The ultimate goal is to create a biomarker panel capable of identifying individuals at high risk of developing sarcopenia even before clinically significant muscle mass loss occurs, enabling a shift to preventive, rather than reactive, management for this group.

The identification of key signaling pathways (TGF-β, mitochondrial biogenesis, satellite cell pool regulation) provides new targets for pharmacological interventions. The upregulation of SASP (senescence-associated secretory phenotype) and the involvement of AP-1 transcription factors (e.g., JUNB) justify the application of senolytics and SASP inhibitors. Maraviroc, a CCR5 inhibitor, demonstrates potential by reducing SASP in human skeletal muscle models [189].

Spatial transcriptomics data reveal the heterogeneity of age-related changes in muscle tissue. Muscle aging is accompanied not by uniform but by localized expression changes that can be spatially mapped to identify ‘vulnerable’ tissue regions. Spatial analysis of human samples showed decreased expression of the MYH2 marker, confirming the vulnerability of type II fibers in aging, while high-resolution expression analysis in model systems identifies distinct fibrotic niches and histologically consistent areas of fibrosis or regeneration, marked by specific signatures [223,224]. These findings support the development of local delivery systems (nanocapsules, tissue-engineered constructs) for targeted intervention on pathological areas.

Research aimed at understanding and restoring neuromuscular junction (NMJ) function, which plays a key role in sarcopenia development, is of particular importance. snRNA-seq studies have identified transcripts (Lrtm1, Pdzrn4, Etv4) enriched in NMJ myonuclei. Overexpression of ETV4 activates the expression of key NMJ genes (Chrne, Chrna1, Musk, Ache, Lrp4), opening prospects for modulating neuromuscular transmission [196]. AAV1.NT-3 (neurotrophin-3) gene therapy demonstrates efficacy in aging models, improving functional parameters and NMJ integrity via activation of the Akt/mTOR pathway [83].

Accumulated preclinical data indicate significant potential for various classes of epigenetic drugs in modulating muscle atrophy. Histone deacetylase inhibitors (Trichostatin A, MS-275, valproic acid) show efficacy in experimental models of muscle atrophy by suppressing the expression of atrophic genes Murf1 and Atrogin-1 [225,226,227]. Inhibition of the lysine-specific demethylase LSD1 prevents glucocorticoid-induced atrophy, opening new avenues for targeted therapy [228].

The future of sarcopenia therapy is linked to the development of personalized therapeutic strategies based on a comprehensive analysis of individual molecular profiles. Integrating multi-omics data with clinical parameters will allow for classifying patients into different endotypes. Such stratification is particularly valuable for optimizing clinical trial design, enabling the formation of patient cohorts with similar pathogenic characteristics and increasing the likelihood of detecting significant therapeutic effects.

Realizing the potential of multi-omics requires overcoming existing methodological challenges. Technical obstacles include standardizing protocols for skeletal muscle sample collection, analytical pipelines for integrating the generated data, and creating reference databases that account for population specificity. From an ethical standpoint, the implementation of NGS screening raises issues related to data privacy and inequality in access to advanced diagnostics. The successful translation of genomic medicine advances into real clinical practice requires consolidated efforts from researchers, clinicians, the pharmaceutical industry, and regulatory bodies to develop standardized protocols and a regulatory framework.

### 6.3. Personalized Medicine Technologies: Application of Artificial Intelligence and Machine Learning for Analyzing Multidimensional Datasets in Sarcopenia

The advancement of personalized medicine is inextricably linked to the implementation of Artificial Intelligence (AI) and Machine Learning (ML) methods, which have significantly improved approaches to the diagnosis, prediction, and therapy of sarcopenia. The evolution of these technologies demonstrates a transition from analyzing individual data types to comprehensive integrative analysis. The overall pipeline for integrating and processing diverse data types through computational approaches is schematically represented in Figure 6.

The initial application of AI technologies in this field focused on solving practical diagnostic tasks using clinical data and medical images. For instance, work by Sachiyo Onishi and colleagues demonstrated the high efficacy of AI for the automated diagnosis of sarcopenia from CT images [229], while Dawei Zhang and colleagues confirmed the feasibility of using AI for accurate segmentation of the rectus femoris muscle in ultrasound images [230]. Concurrently, the application of ML methods to clinical data identified significant markers of sarcopenia, including albumin, C-reactive protein (CRP), folate, and vitamin D [231]. These studies laid the groundwork for the subsequent application of AI and ML to the integrative analysis of multi-omics data in sarcopenia.

The next step involved applying ML algorithms to analyze metabolomic data. In one study aimed at identifying biomarkers of sarcopenia in patients with liver cirrhosis, untargeted metabolomic analysis of serum from 62 patients, combined with three ML algorithms, identified 60 differential metabolites. Pathway enrichment analysis revealed the involvement of glycerophospholipid metabolism, alpha-linolenic acid metabolism, retrograde endocannabinoid signaling, and choline metabolism. Among four potential biomarkers selected by ML methods (N-acetylcarnosine, 2-stearyl citrate, CerP (d18:1/12:0), and 3-methyl-alpha-ionyl acetate), N-acetylcarnosine demonstrated the highest diagnostic accuracy. The obtained data indicate substantial differences in metabolic profiles and open prospects for developing methods for early detection and prediction of sarcopenia as a complication of other diseases [232].

The shift towards analyzing omics data was marked by the development of specialized neural network architectures for processing transcriptomic information. The DSnet-v1 artificial neural network, developed for diagnosing sarcopenia based on gene expression data, is a four-layer deep neural network trained on a carefully selected set of 27 genes from 17,339 initial features [233]. A critically important aspect of this approach is the identification of ethnicity-specific gene expression patterns, underscoring the necessity of accounting for population characteristics when creating diagnostic systems. Additional research in transcriptomics applied ensemble methods, combining Random Forest, XGBoost, and AdaBoost for initial feature selection, followed by training deep neural networks. This approach enabled the identification of key genes associated with FoxO and AMPK signaling pathways, consistent with known mechanisms of muscle atrophy [234].

The advancement of epigenetic data analysis methods led to the creation of diagnostic panels based on DNA methylation patterns and gene expression. The application of LASSO (Least Absolute Shrinkage and Selection Operator) and SVM-RFE (Support Vector Machine–Recursive Feature Elimination) methods allowed for the identification of specific gene sets with increased expression (TPPP3, C1QA, LGR5, MYH8, CDKN1A) and decreased expression (SLC38A1, SERPINA5, HOXB2), providing a molecular basis for understanding the pathogenetic mechanisms of sarcopenia [235].

The most promising direction is the development of methods for the integrative analysis of multiple omics platforms. The Multi-Task Attention-aware method for Multi-Omics data (MTA-MO) represents a revolutionary approach to identifying drug targets and drug repurposing. Applying MTA-MO to multi-omics data identified seven potential hub genes associated with sarcopenia: ESR1, ATM, CDC42, EP300, PIK3CA, EGF, and PTK2B. A significant achievement was the identification of canagliflozin as a promising candidate for repurposing, demonstrating high binding affinity for the PTK2B protein [236].

In parallel, methods of Artificial Neural Network Inference (ANNi), based on a swarm neural network integrated into a deep learning model, are being developed. This approach identified CHAD, ZDBF2, USP54, and JAK2 as genes having the strongest interactions predictive of muscle aging. Experimental validation confirmed significant upregulation of USP54, CHAD, and ZDBF2 in aging muscles, and functional analysis revealed enrichment of signaling pathways related to bone development, immune response, and apoptosis, consistent with known mechanisms of sarcopenia [177]. The corresponding artificial intelligence methods applied to different data types, along with their key outcomes, are systematically summarized in Table 4.

The integration of these diverse artificial intelligence approaches creates a comprehensive foundation for transitioning to personalized management of sarcopenia. Computational methods not only enable the identification of new targets and diagnostic biomarkers but also unravel the complex networks of molecular interactions underlying sarcopenia. Combining multi-omics research data with clinical parameters opens possibilities for developing predictive risk models and personalized therapeutic strategies. Future prospects are linked to the creation of more sophisticated integrative models that combine multi-omics data with clinical and functional parameters. These technologies will allow not only for processing unprecedented volumes of multidimensional data but also for uncovering complex, non-linear relationships that remain unnoticed when analyzing individual data layers. The further development of AI approaches in close integration with fundamental and clinical research will contribute to establishing a new paradigm for the diagnosis, prevention, and treatment of sarcopenia based on the principles of precision medicine.

## 7. Clinical Implications and Translational Outlook

Disruption of the neuromuscular junction is a key mechanism in age-related muscle decline and presents a tractable target for diagnostic and therapeutic development. The serum C-terminal agrin fragment reflects NMJ disassembly and correlates with sarcopenia, supporting its utility for screening and longitudinal monitoring in older adults [220,237,238].

The Motor Unit Number Index, an electrophysiological measure, captures lower motor neuron loss and defines a neurogenic endotype. This may inform patient stratification, guide therapeutic interventions, and serve as a pharmacodynamic biomarker [239,240].

Exercise interventions incorporating progressive resistance and endurance training have been shown to stabilize NMJs and promote remodeling via PGC-1α activation. These programs offer a low-risk, first-line approach and a platform for combination therapies [241,242].

Pharmacologic strategies such as myostatin or ActRII inhibition consistently increase lean mass. However, functional gains are variable, indicating a need for patient enrichment using NMJ-focused biomarkers and composite outcome measures [243,244].

Senescence-targeting approaches show preclinical efficacy in enhancing muscle adaptation and hypertrophy but require biomarker-driven clinical trials with rigorous safety monitoring [245].

Gene therapy using AAV1.NT-3 has demonstrated improvements in both muscle strength and NMJ integrity in aged mice, supporting first-in-human trials incorporating CAF and MUNIX as markers of target engagement [246].

Multi-omics profiling of aging skeletal muscle has identified CCR5 antagonism—specifically with maraviroc—as a candidate senotherapeutic intervention that warrants controlled clinical evaluation [189].

Collectively, these findings support a translational roadmap that integrates CAF and MUNIX for patient selection and monitoring, combines structured exercise with mechanism-targeted agents in adaptive trial designs, and anchors clinical efficacy to mechanistic biomarkers to reduce translational risk and accelerate therapeutic impact.

## 8. Conclusions

This review provides a comprehensive analysis of the current understanding of the neurogenic component in sarcopenia development, based on data obtained using high-throughput sequencing technologies. The systematic analysis conducted demonstrates a paradigm shift in the understanding of sarcopenia pathogenesis—from traditional views of primary muscle dysfunction to the recognition of neuromuscular dysregulation as a key initiating and sustaining mechanism of this pathological condition.

The integration of genomic, transcriptomic, and epigenomic findings has enabled the formulation of a unified molecular model of neurogenic sarcopenia, centered on a cascade of interconnected pathological processes initiated by age-related destabilization of the neuromuscular junction. NGS technologies have revealed the complex architecture of genetic predisposition to sarcopenia, encompassing both rare, high-penetrance variants in genes of neuromuscular function and polygenic risk factors. Epigenetic analysis has demonstrated the existence of interconnected pathogenic axes of dysregulation, expanding the understanding of the disease’s molecular mechanisms beyond traditional concepts.

The revolutionary contribution of single-cell analysis lies in demonstrating sarcopenia as a systemic disease of the muscle niche, characterized by coordinated changes across all resident cell populations. The identification of muscle stem cell dysfunction, the accumulation of senescent cells with a pro-inflammatory secretory phenotype, and impaired intercellular communication fosters a new understanding of pathogenesis as a result of systemic disorganization of tissue homeostasis.

The accumulated scientific data provide a solid foundation for developing personalized therapeutic strategies. The identification of key molecular targets, including epigenetic regulators, neurotrophic factors, and components of the neuromuscular junction, opens avenues for creating targeted therapeutic approaches. The concept of repurposing existing drugs based on transcriptomic signatures and sex-specific differences in molecular profiles is of particular significance. The advancement of experimental models, including vascularized organoids and innervated muscle constructs, provides a platform for precise modeling of pathophysiological processes and testing therapeutic strategies.

The integration of these approaches with artificial intelligence and machine learning technologies enables the creation of predictive models for disease development and progression. The application of deep learning algorithms to integrate genomic, transcriptomic, proteomic, and metabolomic data facilitates the identification of complex, non-linear relationships that remain inaccessible to traditional analytical methods.

Despite significant progress, fundamental challenges remain to be addressed for the full realization of genomic medicine’s potential in the context of sarcopenia. Technical limitations include the need for standardized protocols, analytical pipelines, and the creation of population-specific reference databases.

Priority directions for future work include the functional validation of identified genetic and epigenetic targets using advanced experimental models, the development of multi-omics diagnostic panels for patient stratification and therapy response prediction, and the creation of integrative computational models that combine molecular data with clinical and functional parameters. The neurogenic nature of sarcopenia, confirmed by convergent data from various omics platforms, positions this disease within the continuum of neurodegenerative conditions and opens possibilities for applying therapeutic strategies developed for other neurological disorders.

In conclusion, it is essential to emphasize that neurogenic sarcopenia represents not merely a medical problem but also a unique model for studying the aging processes of the neuromuscular system. Understanding the mechanisms underlying age-related denervation and muscle atrophy is fundamentally important not only for developing therapeutic approaches to sarcopenia but also for the broader comprehension of aging biology and the development of strategies for healthy longevity.

## Figures and Tables

**Figure 1 ijms-26-11185-f001:**
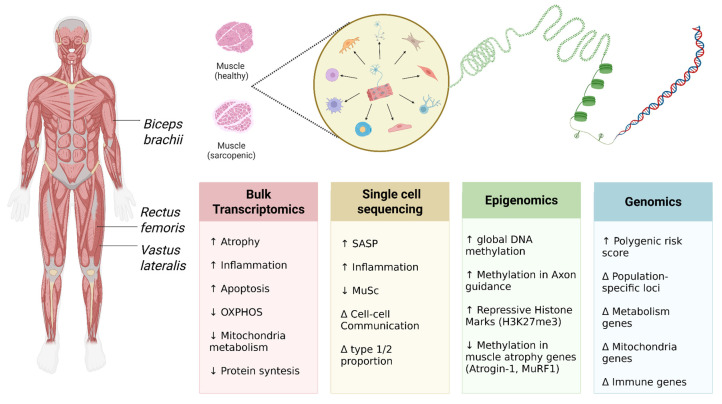
Multiscale NGS Landscape of Neurogenic Sarcopenia. This schematic provides an overview of NGS-based strategies for elucidating the molecular mechanisms of sarcopenia. The analysis focuses on key skeletal muscles frequently subjected to biopsy in clinical studies: the vastus lateralis, rectus femoris, and biceps brachii. Transcriptomic analysis (bulk RNA-seq) has revealed global gene expression alterations at the tissue level. Single-cell sequencing has enabled the identification of cell-specific transcriptomic profiles and disruptions in intercellular communication. Epigenomic studies have demonstrated alterations in DNA methylation patterns and histone modifications in genomic regions regulating muscle tissue homeostasis. Genomic analysis has identified pathogenic and protective genetic variants in genes associated with impaired neuromuscular axis function. The symbol △ denotes identified alterations/changes in the specified parameter.

**Figure 2 ijms-26-11185-f002:**
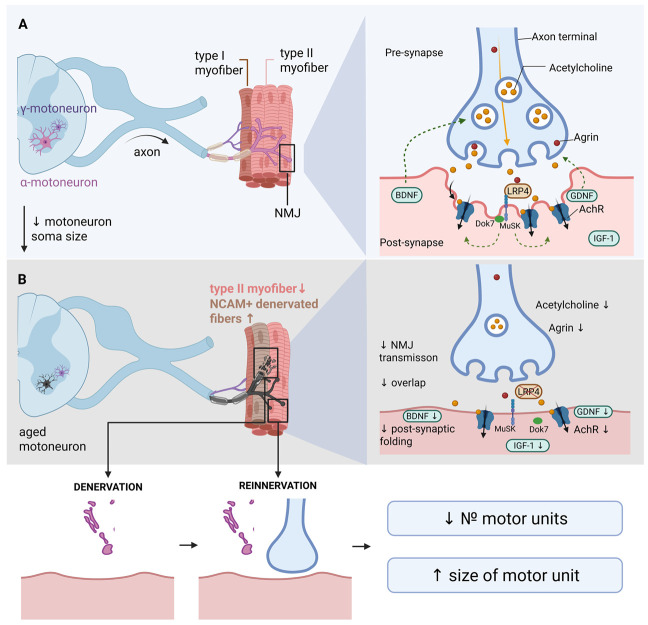
Age-related changes in the motoneuron and neuromuscular junction: a comparative analysis of a healthy and an aging neuromuscular axis. (**A**) Healthy Motoneuron and NMJ. The motoneuron is characterized by a large cell body and a well-developed dendritic network, ensuring efficient reception of synaptic signals. The presynaptic terminal contains a high concentration of acetylcholine vesicles, facilitating efficient neurotransmitter release into the synaptic cleft. The postsynaptic membrane exhibits dense, organized clustering of nicotinic acetylcholine receptors (nAChRs), ensuring reliable signal transmission. Synaptic function is supported by optimal expression levels of neurotrophic factors (BDNF, GDNF and IGF-1), which regulate the plasticity and stability of the junction. A key element is the robust functioning of the Agrin-Lrp4-MuSK-DOK7 signaling cascade, which ensures the formation and maintenance of receptor clusters on the postsynaptic membrane, guaranteeing highly efficient neuromuscular transmission. (**B**) Aging Motoneuron and NMJ. The motoneuron cell body shows signs of atrophy, including a reduction in soma size and a simplification of the dendritic arbor, alongside areas of axonal demyelination. In response to progressive denervation, manifested by partial loss of synaptic contacts and significant fragmentation of neuromuscular synapses, surviving axons initiate a process of compensatory collateral sprouting—the formation of new terminal branches. At the molecular level, there is a significant reduction in the synthesis and release of acetylcholine, correlating with a decreased number of synaptic vesicles in the presynaptic terminal. The postsynaptic membrane is characterized by a critical decrease in nAChR density and disruption of their clustered organization. These changes are exacerbated by reduced expression of key neurotrophic factors and progressive dysfunction of the Agrin-Lrp4-MuSK-DOK7 axis, leading to impaired synaptic stability.

**Figure 3 ijms-26-11185-f003:**
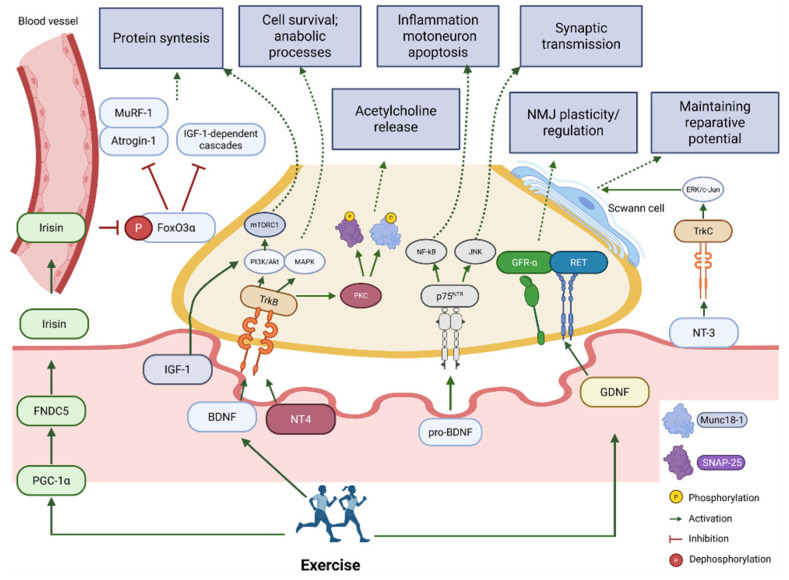
Schematic representation of the neurotrophic network regulating neuromuscular junction (NMJ) homeostasis in sarcopenia. The figure illustrates key neurotrophic factors, their receptors, and intracellular signaling cascades impacting the presynaptic nerve terminal, postsynaptic muscle membrane, satellite cells, and muscle fiber metabolism. An imbalance within this network, characterized by age-related alterations in the ratios of factors and their receptors, leads to NMJ destabilization, motor neuron loss, and muscle atrophy.

**Figure 4 ijms-26-11185-f004:**
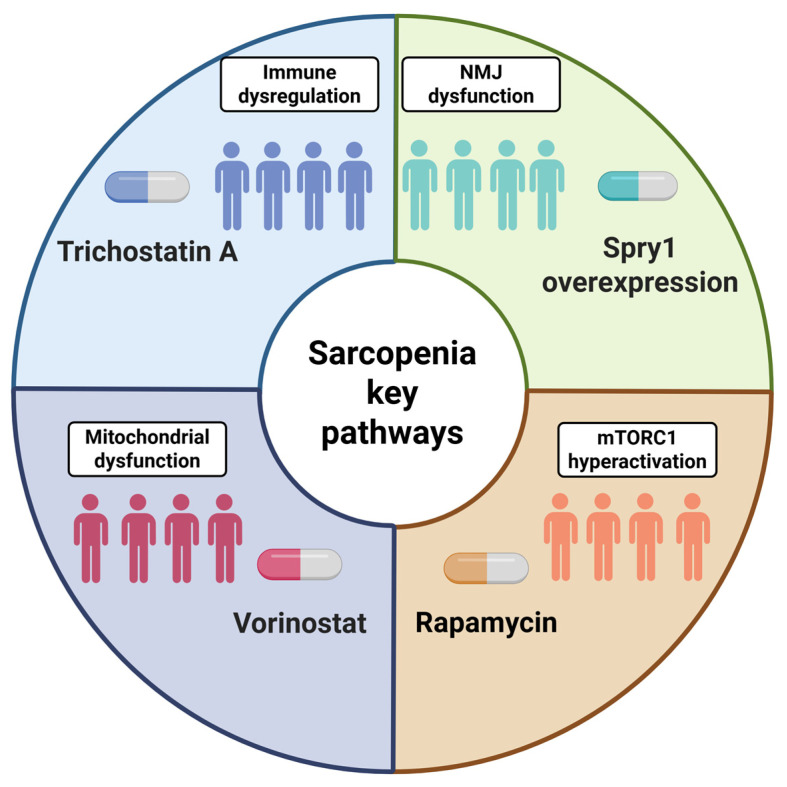
Conceptual framework for sarcopenia intervention. The diagram visualizes a paradigm for combating sarcopenia by targeting its central drivers. For each dysregulated process, candidate corrective agents are shown, illustrating the translation of molecular insights into potential clinical applications. This framework provides a rationale for drug repurposing and the development of novel therapeutics aimed at specific nodes of the sarcopenia network.

**Figure 5 ijms-26-11185-f005:**
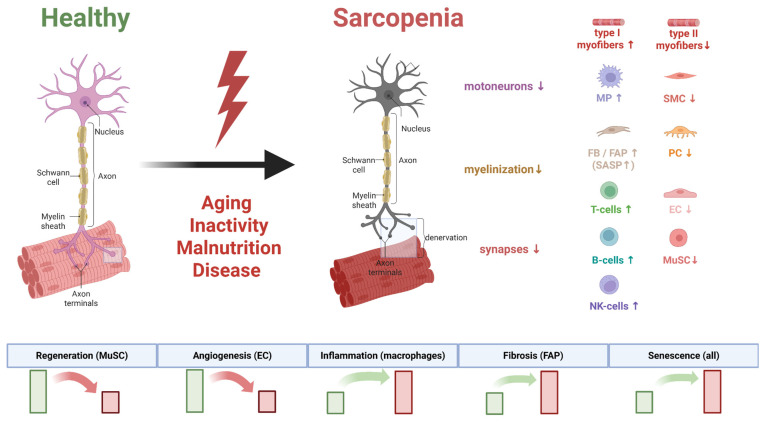
Pathological Remodeling of the Neuromuscular Junction (NMJ) in Sarcopenia. This schematic illustrates the transition from a healthy NMJ to a dysfunctional one characteristic of sarcopenia. A lightning bolt symbolizes the accelerated degenerative processes driven by key risk factors: Aging, Inactivity, Malnutrition, and Underlying Disease. The schematic on the right details the shifts in cellular populations within the neuromuscular axis, highlighting which populations increase or decrease in the context of sarcopenia. The bar graph quantifies critical tissue-level processes: a decrease in Regeneration (driven by Muscle Stem Cells–MuSCs) and Angiogenesis (Endothelial Cells–ECs), alongside an increase in Inflammation (Macrophages), Fibrosis (Fibro-Adipogenic Progenitors–FAPs), and overall cellular Senescence. The diagram also features other stromal cells that contribute to tissue dysfunction: fibroblasts (FBs), pericytes (PCs) and smooth muscle cells (SMCs), which play a key role in maintaining muscle homeostasis.

**Figure 6 ijms-26-11185-f006:**
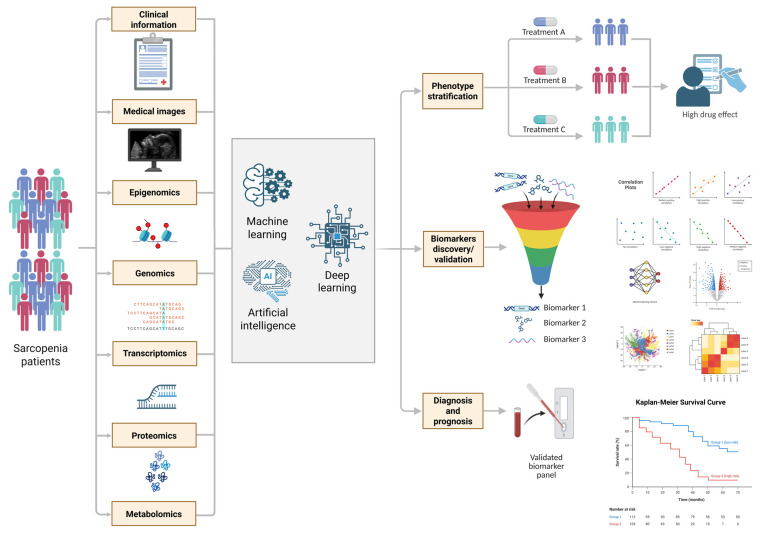
A unified computational framework for sarcopenia research using artificial intelligence. The schematic illustrates the integrative pipeline for sarcopenia analysis, where multi-omics data, clinical parameters, and medical imaging are processed through a suite of machine learning (ML) and deep learning (DL) algorithms. This computational integration enables the derivation of actionable insights for patient stratification, biomarker identification, and enhanced diagnostic and prognostic precision.

**Table 1 ijms-26-11185-t001:** Role of Neurotrophic Factors in the Development of Sarcopenia.

Factor	Mechanism	Role
BDNF	Signaling via TrkB and p75NTR receptors; modifies the presynaptic exocytotic machinery (phosphorylation of Munc18-1, SNAP-25) and influences downstream kinases (PKC-βI/PKCε)	Involved in maintaining NMJ integrity and function; altered levels are associated with sarcopenia markers in humans and age-related NMJ changes in animal models
NT-4	Shares the receptor pathway with BDNF via TrkB and p75NTR, influencing presynaptic exocytosis and the activity of PKC-dependent pathways	A postsynaptically released factor involved in neuroprotection and regulation of neurotransmission at the NMJ; an altered BDNF/NT-4 ratio is noted with aging
NGF	Acts through high-affinity TrkA and p75NTR receptors; supports neuronal survival and differentiation	Influences neuronal survival and motoneuron status during aging; identified as a positive regulator of muscle mass
NT-3	Acts through high-affinity TrkC/TrkA and p75NTR receptors; supports neuronal survival and differentiation	
GDNF	Acts via the Ret/GFRα receptor system, stimulating pro-survival neuronal signals and maintaining synaptic stability	Supports motoneuron survival and helps preserve innervation; implicated in neuromuscular plasticity
NRG4	Adipokine-like action; improves insulin sensitivity and indirectly contributes to the preservation of muscle mass	Levels decrease with age; administration of recombinant NRG4 in aged mice improves signs of sarcopenia and metabolic disorders
IGF-1	Activation of IGF-1R → PI3K/AKT mechanisms stimulates protein synthesis and suppresses catabolic pathways	Anabolic factor promoting muscle cell growth and survival; protects against age-related loss of mass and strength
Irisin	Secreted myokine with anti-inflammatory and neuroprotective effects; modulates mitochondrial function and metabolism	Positive regulator of muscle function; its levels correlate with physical performance in the elderly

**Table 2 ijms-26-11185-t002:** Pathogenic mechanisms linking sarcopenia with Alzheimer’s and Parkinson’s diseases.

Link to Parkinson’s Disease (PD)	Link to Alzheimer’s Disease (AD)	Specific Processes and Manifestations	Pathogenic Mechanism
Pathological α-synuclein aggregation in motor neuron axons and NMJs, directly disrupting their structure and function, leading to denervation.	Altered NMJ structure (fragmentation, partial denervation) in AD models (3xTgAD mice). Accumulation of APP and β-amyloid in muscle tissue.	Muscle fiber denervation, NMJ degeneration, motor neuron dysfunction.	Neurogenic Mechanisms and Neuromuscular Junction (NMJ) Dysfunction
α-Synuclein impairs the mitochondrial importer TOM40. Decreased expression of oxidative phosphorylation genes in muscle. Sharp increase in ROS production.	Disrupted neuronal energy homeostasis, ROS accumulation. Mitochondrial dysfunction in skeletal muscle leads to reduced strength and apoptosis activation.	Disrupted dynamics, biogenesis, and mitophagy; accumulation of mtDNA damage; reduced energy production.	Mitochondrial Dysfunction
Oxidative stress and inflammation create a self-sustaining pathological cycle that enhances α-synuclein aggregation and muscle tissue damage.	Pro-inflammatory mediators in AD exert systemic effects on skeletal muscle, creating a vicious cycle that exacerbates sarcopenia.	Chronic low-grade inflammation (inflammaging); release of pro-inflammatory cytokines.	Neuroinflammation/Systemic Inflammation
Key role of α-synuclein: its aggregates in NMJs impair acetylcholine release and increase abnormal mitochondria count.	Accumulation of β-amyloid and APP in muscle tissue, contributing to dysfunction.	Accumulation and aggregation of specific pathological proteins with direct toxic effects.	Pathological Protein Aggregates
α-Synuclein reduces basal respiratory and glycolytic capacity of MuSCs, impairing their migration and fusion, critically compromising muscle regeneration.	May be a consequence of the general neurogenic and inflammatory context.	Dysfunction of muscle satellite cells (MuSCs), impaired metabolism, proliferation, and fusion capacity.	Impaired Regenerative Potential
*NA*	Identified as a key mechanism of age-related motor neuron loss. Physical activity can modulate this process.	Loss of nuclear membrane integrity, increased permeability, accumulation of toxic proteins in the nucleus.	Impaired Nucleocytoplasmic Transport

**Table 3 ijms-26-11185-t003:** Genetic determinants of sarcopenia: summary of WES, GWAS, and WGS studies.

Notes/Replication	Phenotype Assessed	Biological Function/Pathway	Key Variant(s)/Gene(s)	Design and Sample Size	Study/Population
Sub-GWS; replicated in UK Biobank	WLBM	Ubiquitin–proteasome; lipid metabolism	rs740681 (*FZR1*), *SOAT2*	WES	Han Chinese (n = 101) [137]
Significant in discovery, nominal in replication	WLBM	Actin signaling, glycoprotein biosynthesis, ATPase	rs3732593 (3p27.1; *MCF2L2*, *B3GNT5*, *ATP11B*)	GWAS	Framingham Heart Study (n = 6004) [138,139]
PRS ≥ 4 alleles: very high OR (630.6)	Clinical diagnosis	Cholesterol binding, apoptosis	rs10282247 (*OSBPL3*), rs7022373 (*ACER2*)	GWAS (clin. sarcopenia)	Taiwanese elderly [140]
eQTL confirmed expression in skeletal muscle	LBM, ASM	mRNA destabilization, immune signaling	rs1187118, rs3768582 (*RPS10*, *NUDT3*, *NCF2*, *SMG7*, *ARPC5*)	GWAS meta-analysis	Korean cohorts (n = 6961) [141]
20 novel loci identified	Grip strength, walking speed	Glycogen synthesis, myogenesis, Ca^2+^ homeostasis	*PPP1R3A*, *ZBTB38*, *ATP2A1*	MTAG + TWAS	UK Biobank (n = 217,822) [145]
Partial overlap with strength traits	Weakness (clinical)	Immune regulation, TGF-β signaling	*HLA-DQA1*, *GDF5*	GWAS	Muscle weakness GWAS (n = 256,523) [146]
Functional validation in model organisms	Muscle mass	Sex-specific loci; mitochondrial/structural pathways	*LINC01661/PRMT6*, *RCC2P8/COL25A1*, *DMAC1*, *EMP2*, *SSUH2*	WGS	Multi-ancestry WGS (n = 10,729) [147]
Protective allele in NE Asians	Sarcopenia diagnosis	Myogenic differentiation, immune protection	*SLC41A3*, *HLA-DPB1*02:01	WGS	Japanese cohort (n = 129) [148]
OR ≈ 1.98; physical activity mitigates risk	Sarcopenia risk	Lipid metabolism, cytoskeleton, signaling	*FADS2*, *MYO10*, *KCNQ5*, *DOCK5*, *LRP1B*	PRS	Korean case–control (1368 vs. 15,472) [149]

**Table 4 ijms-26-11185-t004:** Summary of artificial intelligence methodologies and their applications across data types in sarcopenia research.

Significance and Future Directions	AI/ML Methods	Data Type
Laid the groundwork for diagnostic automation and identification of clinical predictors of sarcopenia	Deep learning architectures (U-Net, nnU-Net, AutoSAM); semantic segmentation (DeepLabV3+ with EfficientNetV2-L); image classification (EfficientNetV2-L); ensemble methods (Random Forest)	Clinical Data and Medical Images
Opened prospects for early detection and prediction of sarcopenia as a complication of other diseases based on metabolic profiles	LASSO, SVM-RFE и RF	Metabolomic Data
Highlighted the necessity of accounting for population characteristics. Deepened understanding of molecular mechanisms via key signaling pathways	Specialized neural network architectures (DSnet-v1); ensemble methods (Random Forest, XGBoost, AdaBoost) followed by deep neural network training	Transcriptomic Data
Provided a molecular basis for understanding the pathogenetic mechanisms of sarcopenia	LASSO, SVM-RFE	Epigenetic Data
The most promising direction. Enables identification of novel drug targets and repurposing candidates, unraveling complex molecular interaction networks	MTA-MO (Multi-Task Attention-aware for Multi-Omics data); ANNi (Artificial Neural Network Inference)	Integrative Multi-Omics Data

## Data Availability

The original contributions presented in this study are included in the article. Further inquiries can be directed to the corresponding author.

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
