# Peer review of "Neural Cues and Genomic Clues: NGS Insights into Neurogenic Sarcopenia and Muscle Atrophy"

_ijms, 2025, doi:10.3390/ijms262211185_

Round 1
Reviewer 1 Report
Comments and Suggestions for Authors
The manuscript is an exceptionally comprehensive and scientifically rigorous work that presents the neurogenic mechanisms of sarcopenia and the role of omics technologies with a modern and integrative perspective. The text is logically structured, up to date, and highly informative. It would be valuable to include a brief Clinical implications / translational outlook Section summarizing how the described mechanisms could be applied in diagnostics or targeted therapeutic development. Regarding the mentioned “innervated organoid” models, what specific advantages or limitations do the authors see in studying human sarcopenia with these systems?
Over what time frame and by what approaches do the authors consider it realistic that modulation of the discussed molecular pathways (eg TrkB, IGF-1R, PGC-1α) could become a therapeutic target? To what extent are the expression and effects of neurotrophic factors influenced by sex, age, or physical activity? These aspects would be worth briefly addressing. Among the proposed biomarkers (BDNF, IGF-1, irisin, etc) which seem most suitable for clinical validation?
Overall this is an excellent modern, and valuable study that significantly contributes to a deeper understanding of the pathogenesis of sarcopenia. The references are relevant, and the work shows a low plagiarism index. I recommend the manuscript for publication.
Author Response
Response to reviewers’ comments on the manuscript by Darya Kupriyanova entitled “Neural Cues and Genomic Clues: NGS Insights into Neurogenic Sarcopenia and muscle atrophy.”
General Comment:
The manuscript is an exceptionally comprehensive and scientifically rigorous work that presents the neurogenic mechanisms of sarcopenia and the role of omics technologies with a modern and integrative perspective. The text is logically structured, up to date, and highly informative.
Response:
Thank you very much for your positive evaluation and encouraging feedback. We sincerely appreciate your recognition of the integrative approach and scientific rigor of our work. Your comments reinforce our confidence that highlighting the neurogenic mechanisms of sarcopenia through multi-omics integration represents a valuable contribution to understanding this multifactorial condition.
Comment 1:
The text is logically structured, up to date, and highly informative. It would be valuable to include a brief Clinical implications / translational outlook Section summarizing how the described mechanisms could be applied in diagnostics or targeted therapeutic development.
Response:
We thank the reviewer for this helpful suggestion. We added a new subsection titled ‘Clinical Implications and Translational Outlook’ at the end of the Discussion, summarizing diagnostic, monitoring, and therapeutic avenues and aligning them with biomarker-anchored trial design. (Lines1592-1622)
Comment 2:
Regarding the mentioned “innervated organoid” models, what specific advantages or limitations do the authors see in studying human sarcopenia with these systems?
Response:
We thank the reviewer for highlighting the relevance of human innervated neuromuscular organoids and NMJ-on-chip systems in sarcopenia research. We fully agree that these platforms offer several key advantages. Notably, they enable de novo formation of neuromuscular junctions with appropriate maturation of the ε-subunit-containing acetylcholine receptors and support functional assays such as calcium imaging, evoked contractions, optogenetic pacing, and force measurement. These capabilities make them highly suitable for mechanism-driven screening efforts and pharmacodynamic biomarker development within a human genetic context [PMID: 31084710, 31956040, 30324134, 33382092].
At the same time, we acknowledge several important limitations. The process of induced pluripotency resets epigenetic age, potentially diminishing the expression of late-life phenotypes unless modified by pro-aging stimuli or partial reprogramming strategies [PMID: 27941802, 36871150]. In addition, many current platforms lack vascular, endocrine, and immune components that critically influence inflammaging, denervation, and reinnervation dynamics—though progress in vascularized and immune-inclusive co-culture systems is ongoing [PMID: 33912545, 36716724]. Mechanical loading and hormonal cues are often simplified, limiting the generalizability of findings to real-world contexts involving mobility, nutrition, and comorbidities. Furthermore, muscle fibers in these systems frequently show incomplete maturation, including immature fiber-type specification and underdeveloped myonuclear domains.
Taken together, we view innervated human neuromuscular organoids as valuable tools for hypothesis generation and preclinical prioritization. However, we emphasize the importance of careful endpoint selection and validation against clinical data from aging human cohorts to ensure translational relevance.
Comment 3:
Over what time frame and by what approaches do the authors consider it realistic that modulation of the discussed molecular pathways (eg TrkB, IGF-1R, PGC-1α) could become a therapeutic target? To what extent are the expression and effects of neurotrophic factors influenced by sex, age, or physical activity? These aspects would be worth briefly addressing. Among the proposed biomarkers (BDNF, IGF-1, irisin, etc) which seem most suitable for clinical validation?
Response:
We appreciate this constructive suggestion and agree that pathway-specific modulation offers a promising therapeutic framework. In our view, realistic implementation aligns across three translational horizons:
Near-term (1–3 years):
Structured resistance–endurance exercise programs remain the most immediate and scalable intervention, with well-established effects on PGC-1α and BDNF signaling. Exercise robustly induces mitochondrial biogenesis and elevates peripheral BDNF levels, supporting the use of biomarker-anchored trials in older adults [PMID: 9014996, 4314337, 40459444, 546851; summarized in 23050972].
Mid-term (3–5 years):
Combination trials that enrich for individuals likely to benefit from interventions targeting the IGF-1 axis (e.g., GH/IGF-1 modulation) represent a logical next step. Exercise consistently increases circulating IGF-1 levels in frail and sarcopenic older adults, and multiple studies link IGF-1 signaling to sarcopenia risk through genetic and observational data [PMID: 40917423, 7140321, 1422472].
Longer-term (5–10+ years):
Gene therapy strategies such as AAV1.NT-3 delivery offer a future avenue for trophic support. In aged mice, this approach improves NMJ structure and muscle function, though clinical translation will require careful evaluation using NMJ-anchored pharmacodynamic markers [PMID: 36897179]. Small-molecule TrkB agonists remain at the preclinical stage, with unresolved challenges around pharmacokinetics and receptor specificity [PMID: 10479861, 29500536, 21900882], placing them on a longer developmental timeline.
On modulators of neurotrophic factor biology:
Sex and age effects: Plasma BDNF declines with age and displays sex hormone–linked variation. Given the confounding contribution of platelet-derived BDNF in serum, we emphasize the need for plasma or platelet-poor plasma in clinical assays [PMID: 15585351, 5271179, 11577193]. Similarly, IGF-1 levels decrease with age and are modulated by sex hormones and nutritional status [PMID: 12234294].
Physical activity: Regular exercise elevates plasma BDNF and increases serum IGF-1 levels in frail older adults. However, the effects on circulating irisin are variable across studies and cohorts [PMID: 9014996, 4314337, 40917423, 12904].
On candidate circulating biomarkers:
IGF-1, particularly when paired with IGFBP-3 or assessed as an IGF-1/IGFBP-3 ratio, is well-suited for near-term clinical validation due to assay standardization and established links to aging-related outcomes [PMID: 9640414, 7140321].
BDNF is a promising dynamic marker for target engagement, but pre-analytic considerations are critical—plasma or platelet-poor plasma should be used to avoid artifactual elevations from platelet release [PMID: 11577193, 17989].
Irisin lacks standardized immunoassays, and existing data are inconsistent. If pursued, quantification should rely on validated LC-MS/MS protocols. At this stage, we consider it a lower-priority biomarker pending further assay harmonization [PMID: 26278051, 7033458, 12904].
Reviewer 2 Report
Comments and Suggestions for Authors
This is a well-written and comprehensive review article with adequate novelty. Some points should be addressed.
- The Abstract should be re-organized and distinctly included background/objectives, methods, results and conclusions sections.
- The authors should include more references in the 2nd paragraph of the Introduction section.
- Again, the authors should include more references in the paragraph of lines 88-112 of the Introduction section. This paragraph is quite long and it could be split into two smaller paragraphs.
- The authors should emphasize at the last paragraph of the Introduction what literature gap cover their review article.
- After the Introduction section, the authors should add a Methods section describing how the collect their data (What database and keywords used? Are there any inclusion and exclusion criteria?, etc).
- A figure in section 2.2 could be very useful for the readers.
- A table in section 2.3 could be very useful for the readers.
- A figure in section 4.1 could be very useful for the readers.
- A table in section 5.3 could be very useful for the readers.
Author Response
Response to reviewers’ comments on the manuscript by Darya Kupriyanova entitled “Neural Cues and Genomic Clues: NGS Insights into Neurogenic Sarcopenia and muscle atrophy.”
General Comment:
This is a well-written and comprehensive review article with adequate novelty. Some points should be addressed.
Response:
Thank you very much for your positive evaluation and encouraging feedback. We sincerely appreciate your recognition of the integrative approach and scientific rigor of our work. Your comments reinforce our confidence that highlighting the neurogenic mechanisms of sarcopenia through multi-omics integration represents a valuable contribution to understanding this multifactorial condition.
Comment 1:
The Abstract should be re-organized and distinctly included background/objectives, methods, results and conclusions sections.
Response:
We thank the reviewer for this valuable recommendation. The Abstract has been fully reorganized into four clearly defined sections — Background, Objectives, Methods, and Conclusions — in accordance with the journal’s structure. This revision improves readability and transparency, ensuring that the scope, methodological framework, and key outcomes of the review are presented in a concise and systematic manner. (lines 19-47).
Comment 2:
The authors should include more references in the 2nd paragraph of the Introduction section.
Response:
We appreciate the reviewer’s comment. Additional references have been incorporated into the second paragraph of the Introduction to strengthen the contextual background. Specifically, we added the following supporting sources: [PMID: 34315158; 37139947; 30993881; 30080217]. These citations expand the evidence base regarding the epidemiological and clinical impact of sarcopenia and provide a more comprehensive overview of its global significance.
Comment 3:
Again, the authors should include more references in the paragraph of lines 88-112 of the Introduction section. This paragraph is quite long and it could be split into two smaller paragraphs
Response:
We thank the reviewer for this constructive suggestion. The paragraph corresponding to lines 88–112 in the Introduction has been divided into two shorter paragraphs to improve readability and logical flow. Additionally, we have incorporated several new references to strengthen the methodological and molecular context, specifically [PMID: 38649488; 35188098; 32478482; 33510174; 29477142; 36516485]. These citations provide broader coverage of next-generation sequencing applications, genetic variant analyses, transcriptomic and single-cell studies relevant to the neurogenic mechanisms of sarcopenia.
Comment 4:
The authors should emphasize at the last paragraph of the Introduction what literature gap cover their review article
Response:
We thank the reviewer for this valuable comment. We have revised the last paragraph of the Introduction to explicitly define the literature gap addressed by our review. The updated paragraph emphasizes that most prior works focus on metabolic and inflammatory mechanisms of sarcopenia, whereas the neurogenic component has not been comprehensively analyzed through multi-omics integration. The new version also clarifies that our review aims to synthesize current NGS-based evidence on structural and functional changes in the nervous system, highlighting how integrative omics and advanced modeling platforms can bridge the gap between molecular discovery and clinical translation.
Comment 5:
After the Introduction section, the authors should add a Methods section describing how the collect their data (What database and keywords used? Are there any inclusion and exclusion criteria?, etc)
Response:
We thank the reviewer for this helpful recommendation. In response, we have added a new Methods section titled Search strategy and selection criteria immediately following the Introduction. This section describes in detail the databases used (PubMed, Web of Science, and Scopus), the time frame of the search, the specific keywords and Boolean operators applied, as well as the inclusion and exclusion criteria for study selection. We also clarify that the review follows a narrative approach, with transparent reporting of screening and synthesis procedures.
Comment 6:
A figure in section 2.2 could be very useful for the readers.
A table in section 2.3 could be very useful for the readers.
A figure in section 4.1 could be very useful for the readers.
A table in section 5.3 could be very useful for the readers.
Response:
We thank the reviewer for these valuable suggestions. In response, we have added the following visual materials to enhance clarity and reader engagement. Due to the inclusion of a new Methods section following the Introduction, the numbering of all subsequent sections has shifted forward by one.
A figure in Section 3.2 illustrating the key mechanisms of neuromuscular junction remodeling in sarcopenia.
A table in Section 3.3 summarizing major omics findings and dysregulated molecular pathways.
A figure in Section 5.1 depicting signaling interactions among TrkB, IGF-1R, and PGC-1α pathways.
A table in Section 6.3 presenting candidate biomarkers with their molecular functions and readiness for clinical validation.
Round 2
Reviewer 2 Report
Comments and Suggestions for Authors
The authors have significantly revised and improved their manuscript.
The abstract is well-written; however, it exceeds the length of that proposed by journal gidelines. The authors should try to reduce it a bit.
Author Response
Response to reviewers’ comments on the manuscript by Darya Kupriyanova entitled “Neural Cues and Genomic Clues: NGS Insights into Neurogenic Sarcopenia and muscle atrophy.”
General Comment:
The abstract is well-written; however, it exceeds the length of that proposed by journal gidelines. The authors should try to reduce it a bit.
Response:
Thank you, we have corrected the abstract.